# Quantile Propagation for Wasserstein-Approximate Gaussian Processes

Rui Zhang[1,2], Christian J. Walder[1,2], Edwin V. Bonilla[2,4], Marian-Andrei Rizoiu[2,3], and Lexing Xie[1,2]

[1]The Australian National University
[2]CSIRO's Data61, Australia
[3]University of Technology Sydney
[4]The University of Sydney
{*rui.zhang, lexing.xie*}*@anu.edu.au*, {*christian.walder, edwin.bonilla*}*@data61.csiro.au*
*marian-andrei.rizoiu@uts.edu.au*

## Abstract

Approximate inference techniques are the cornerstone of probabilistic methods based on Gaussian process priors. Despite this, most work approximately optimizes standard divergence measures such as the Kullback-Leibler (KL) divergence, which lack the basic desiderata for the task at hand, while chiefly offering merely technical convenience. We develop a new approximate inference method for Gaussian process models which overcomes the technical challenges arising from abandoning these convenient divergences. Our method—dubbed Quantile Propagation (QP)—is similar to expectation propagation (EP) but minimizes the $L_2$ Wasserstein distance (WD) instead of the KL divergence. The WD exhibits all the required properties of a distance metric, while respecting the geometry of the underlying sample space. We show that QP matches quantile functions rather than moments as in EP and has the same mean update but a smaller variance update than EP, thereby alleviating EP's tendency to over-estimate posterior variances. Crucially, despite the significant complexity of dealing with the WD, QP has the same favorable locality property as EP, and thereby admits an efficient algorithm. Experiments on classification and Poisson regression show that QP outperforms both EP and variational Bayes.

## 1 Introduction

Gaussian process (GP) models have attracted the attention of the machine learning community due to their flexibility and their capacity to measure uncertainty. They have been widely applied to learning tasks such as regression [32], classification [57, 21] and stochastic point process modeling [38, 62]. However, exact Bayesian inference for GP models is intractable for all but the Gaussian likelihood function. To address this issue, various approximate Bayesian inference methods have been proposed, such as Markov Chain Monte Carlo [MCMC, see *e.g.* 41], the Laplace approximation [57], variational inference [26, 42] and expectation propagation [43, 37].

The existing approach most relevant to this work is expectation propagation (EP), which approximates each non-Gaussian likelihood term with a Gaussian by iteratively minimizing a set of local forward Kullback-Leibler (KL) divergences. As shown by Gelman et al. [17], EP can scale to very large datasets. However, EP is not guaranteed to converge, and is known to over-estimate posterior variances [34, 27, 20] when approximating a short-tailed distribution. By over-estimation, we mean that the approximate variances are larger than the true variances so that more distribution mass lies in the *ineffective* domain. Interestingly, many popular likelihoods for GPs results in short-tailed

posterior distributions, such as Heaviside and probit likelihoods for GP classification and Laplacian, Student's t and Poisson likelihoods for GP regression.

The tendency to over-estimate posterior variances is an inherent drawback of the forward KL divergence for approximate Bayesian inference. More generally, several authors have pointed out that the KL divergence can yield undesirable results such as (but not limited to) over-dispersed or under-dispersed posteriors [11, 30, 22].

As an alternative to the KL, optimal transport metrics—such as the Wasserstein distance [WD, 55, §6]—have seen a recent boost of attention. The WD is a natural distance between two distributions, and has been successfully employed in tasks such as image retrieval [49], text classification [24] and image fusion [7]. Recent work has begun to employ the WD for inference, as in Wasserstein generative adversarial networks [2], Wasserstein variational inference [1] and Wasserstein auto-encoders [54]. In contrast to the KL divergence, the WD is computationally challenging [8], especially in high dimensions [4], in spite of its intuitive formulation and excellent performance,.

**Contributions**. In this work, we develop an efficient approximate Bayesian scheme that minimizes a specific class of WD distances, which we refer to as the $L_2$ WD. Our method overcomes some of the shortcomings of the KL divergence for approximate inference with GP models. Below we detail the three main contributions of this paper.

First, in section 4, we develop quantile propagation (QP), an approximate inference algorithm for models with GP priors and factorized likelihoods. Like EP, QP does not directly minimize global distances between high-dimensional distributions. Instead, QP estimates a fully coupled Gaussian posterior by iteratively minimizing *local* divergences between two particular marginal distributions. As these marginals are univariate, QP boils down to an iterative quantile function matching procedure (rather than moment matching as in EP) — hence we term our inference scheme *quantile propagation*. We derive the updates for the approximate likelihood terms and show that while the QP mean estimates match those of EP, the variance estimates are lower for QP.

Second, in section 5 we show that like EP, QP satisfies the locality property, meaning that it is sufficient to employ *univariate* approximate likelihood terms, and that the updates can thereby be performed efficiently using only the marginal distributions. Consequently, although our method employs a more complex divergence than that of EP ($L_2$ WD vs KL), it has the same computational complexity, after the precomputation of certain (data independent) lookup tables.

Finally, in section 6 we employ eight real-world datasets and compare our method to EP and variational Bayes (VB) on the tasks of binary classification and Poisson regression. We find that in terms of predictive accuracy, QP performs similarly to EP but is superior to VB. In terms of predictive uncertainty, however, we find QP superior to both EP and VB, thereby supporting our claim that QP alleviates variance over-estimation associated with the KL divergence when approximating short-tailed distributions [34, 27, 20].

## 2   Related Work

The basis of the EP algorithm for GP models was first proposed by Opper and Winther [43] and then generalized by Minka [36, 37]. Power EP [33, 34] is an extension of EP that exploits the more general $\alpha$-divergence (with $\alpha = 1$ corresponding to the forward KL divergence in EP) and has been recently used in conjunction with GP pseudo-input approximations [5]. Although generally not guaranteed to converge locally or globally, Power EP uses fixed-point iterations for its local updates and has been shown to perform well in practice for GP regression and classification [5]. In comparison, our approach uses the $L_2$ WD, and like EP, it yields convex local optimizations for GP models with factorized likelihoods. This convexity benefits the convergence of the local update, and is retained even with the general $L_p$ ($p \geq 1$) WD as shown in Theorem 1. Moreover, for the same class of GP models, both EP and our approach have the locality property [50] and can be unified in the generic message passing framework [34].

Without the guarantee of convergence for the explicit global objective function, understanding EP has proven to be a challenging task. As a result, a number of works have instead attempted to directly minimize divergences between the true and approximate joint posteriors, for divergences such as the KL [26, 10], Rényi [30], $\alpha$ [23] and optimal transport divergences [1]. To deal with the infinity issue of the KL (and more generally the Rényi and $\alpha$ divergences) which arises from different

distribution supports [39, 2, 19], Hensman et al. [22] employ the product of tilted distributions as an approximation. A number of variants of EP have also been proposed, including the convergent double loop algorithm [44], parallel EP [35], distributed EP built on partitioned datasets [60, 17], averaged EP assuming that all approximate likelihoods contribute similarly [9], and stochastic EP which may be regarded as sequential averaged EP [29].

The $L_2$ WD between two Gaussian distributions has a closed form expression [12]. Detailed research on the Wasserstein geometry of the Gaussian distribution is conducted by Takatsu [53]. Recently, this closed form expression has been applied to robust Kalman filtering [51] and to the analysis of populations of GPs [31]. A more general extension to elliptically contoured distributions is provided by Gelbrich [16] and used to compute probabilistic word embeddings [40]. A geometric interpretation for the $L_2$ WD between any distributions [3] has already been exploited to develop approximate Bayesian inference schemes [14]. Our approach is based on the $L_2$ WD but does not exploit these closed form expressions; instead we obtain computational efficiency by leveraging the EP framework and using the quantile function form of the $L_2$ WD for univariate distributions. We believe our work paves the way for further practical approaches to WD-based Bayesian inference.

## 3 Prerequisites

### 3.1 Gaussian Process Models

Consider a dataset of $N$ samples $\mathcal{D} = \{\boldsymbol{x}_i, y_i\}_{i=1}^{N}$, where $\boldsymbol{x}_i \in \mathbb{R}^d$ is the input vector and $y_i \in \mathbb{R}$ is the noisy output. Our goal is to establish the mapping from inputs to outputs via a latent function $f : \mathbb{R}^d \to \mathbb{R}$ which is assigned a GP prior. Specifically, assuming a zero-mean GP prior with covariance function $k(\boldsymbol{x}, \boldsymbol{x}'; \boldsymbol{\theta})$, where $\boldsymbol{\theta}$ are the GP hyper-parameters, we have that $p(\boldsymbol{f}) = \mathcal{N}(\boldsymbol{f}|\boldsymbol{0}, K)$, where $\boldsymbol{f} = \{f_i\}_{i=1}^{N}$, with $f_i \equiv f(\boldsymbol{x}_i)$, is the set of latent function values and $K$ is the covariance matrix induced by evaluating the covariance function at every pair of inputs. In this work we use the squared exponential covariance function $k(\boldsymbol{x}, \boldsymbol{x}'; \boldsymbol{\theta}) = \gamma \exp\left[-\sum_{i=1}^{d}(x_i - x_i')^2/(2\alpha_i^2)\right]$, where $\boldsymbol{\theta} = \{\gamma, \alpha_1, \cdots, \alpha_d\}$. For simplicity, we will omit the conditioning on $\boldsymbol{\theta}$ in the rest of this paper.

Along with the prior, we assume a factorized likelihood $p(\boldsymbol{y}|\boldsymbol{f}) = \prod_{i=1}^{N} p(y_i|f_i)$ where $\boldsymbol{y}$ is the set of all outputs. Given the above, the posterior $\boldsymbol{f}$ is expressed as:

$$p(\boldsymbol{f}|\mathcal{D}) = p(\mathcal{D})^{-1} p(\boldsymbol{f}) \prod_{i=1}^{N} p(y_i|f_i),$$

where the normalizer $p(\mathcal{D}) = \int p(\boldsymbol{f}) \prod_{i=1}^{N} p(y_i|f_i) \, d\boldsymbol{f}$ is often analytically intractable. Numerous problems take this form: binary classification [58], single-output regression with Gaussian likelihood [32], Student's-t likelihood [27] or Poisson likelihood [63], and the warped GP [52].

### 3.2 Expectation Propagation

In this section we review the application of EP to the GP models described above. EP deals with the analytical intractability by using Gaussian approximations to the individual non-Gaussian likelihoods:

$$p(y_i|f_i) \approx t_i(f_i) \equiv \widetilde{Z}_i \mathcal{N}(f_i|\widetilde{\mu}_i, \widetilde{\sigma}_i^2).$$

The function $t_i$ is often called the *site function* and is specified by the *site parameters*: the scale $\widetilde{Z}_i$, the mean $\widetilde{\mu}_i$ and the variance $\widetilde{\sigma}_i^2$. Notably, it is sufficient to use univariate site functions given that the local update can be efficiently computed using the marginal distribution only [50]. We refer to this as the *locality property*. Although in this work we employ a more complex $L_2$ WD, our approach retains this property, as we elaborate in subsection 5.2.

Given the site functions, one can approximate the intractable posterior distribution $p(\boldsymbol{f}|\mathcal{D})$ using a Gaussian $q(\boldsymbol{f})$ as below, where conditioning on $\mathcal{D}$ is omitted from $q(\boldsymbol{f})$ for notational convenience:

$$q(\boldsymbol{f}) = q(\mathcal{D})^{-1} p(\boldsymbol{f}) \prod_{i=1}^{N} t_i(f_i) \equiv \mathcal{N}(\boldsymbol{f}|\boldsymbol{\mu}, \Sigma), \quad \boldsymbol{\mu} = \Sigma \widetilde{\Sigma}^{-1} \widetilde{\boldsymbol{\mu}}, \quad \Sigma = (K^{-1} + \widetilde{\Sigma}^{-1})^{-1}, \quad (1)$$

where $\widetilde{\boldsymbol{\mu}}$ is the vector of $\widetilde{\mu}_i$, $\widetilde{\Sigma}$ is diagonal with $\widetilde{\Sigma}_{ii} = \widetilde{\sigma}_i^2$; $\log q(\mathcal{D})$ is the log approximate model evidence expressed as below and further employed to optimize GP hyper-parameters:

$$\boldsymbol{\theta}^\star = \underset{\boldsymbol{\theta}}{\operatorname{argmax}} \log q(\mathcal{D}) = \sum_{i=1}^{N} \log(\widetilde{Z}_i/\sqrt{2\pi}) - \frac{1}{2}\log|K+\widetilde{\Sigma}| - \frac{1}{2}\widetilde{\boldsymbol{\mu}}^{\mathsf{T}}(K+\widetilde{\Sigma})^{-1}\widetilde{\boldsymbol{\mu}}. \qquad (2)$$

The core of EP is to optimize site functions $\{t_i(f_i)\}_{i=1}^{N}$. Ideally, one would seek to minimize the global KL divergence between the true and approximate posterior distributions $\mathrm{KL}(p(\boldsymbol{f}|\mathcal{D})\|q(\boldsymbol{f}))$, however this is intractable. Instead, EP is built based on the assumption that the global divergence can be approximated by the local divergence $\mathrm{KL}(\widetilde{q}(\boldsymbol{f})\|q(\boldsymbol{f}))$, where $\widetilde{q}(\boldsymbol{f}) \propto q^{\backslash i}(\boldsymbol{f})p(y_i|f_i)$ and $q^{\backslash i}(\boldsymbol{f}) \propto q(\boldsymbol{f})/t_i(f_i)$ are refered to as the tilted and cavity distributions, respectively. Note that the cavity distribution is Gaussian while the tilted distribution is usually not. The local divergence can be simplified from multi-dimensional to univariate, $\mathrm{KL}(\widetilde{q}(\boldsymbol{f})\|q(\boldsymbol{f})) = \mathrm{KL}(\widetilde{q}(f_i)\|q(f_i))$ (detailed in Appendix G), and $t_i(f_i)$ is optimized by minimizing it.

The minimization is realized by projecting the tilted distribution $\widetilde{q}(f_i)$ onto the Gaussian family, with the projected Gaussian denoted $\mathrm{proj}_{\mathrm{KL}}(\widetilde{q}(f_i)) \equiv \operatorname{argmin}_{\mathcal{N}} \mathrm{KL}(\widetilde{q}(f_i)\|\mathcal{N}(f_i))$. Then the projected Gaussian is used to update $t_i(f_i) \propto \mathrm{proj}_{\mathrm{KL}}(\widetilde{q}(f_i))/q^{\backslash i}(f_i)$. The mean and the variance of $\mathrm{proj}_{\mathrm{KL}}(\widetilde{q}(f_i)) \equiv \mathcal{N}(\mu^\star, \sigma^{\star 2})$ match the moments of $\widetilde{q}(f_i)$ and are used to update $t_i(f_i)$'s parameters:

$$\mu^\star = \mu_{\widetilde{q}_i}, \quad \sigma^{\star 2} = \sigma_{\widetilde{q}_i}^2, \qquad (3)$$

$$\widetilde{\mu}_i = \widetilde{\sigma}_i^2 \left( \mu^\star (\sigma^\star)^{-2} - \mu_{\backslash i} \sigma_{\backslash i}^{-2} \right), \quad \widetilde{\sigma}_i^{-2} = (\sigma^\star)^{-2} - \sigma_{\backslash i}^{-2}, \qquad (4)$$

where $\mu_{\widetilde{q}_i}$ and $\sigma_{\widetilde{q}_i}^2$ are the mean and the variance of $\widetilde{q}(f_i)$, and $\mu_{\backslash i}$ and $\sigma_{\backslash i}^2$ are the mean and the variance of $q^{\backslash i}(f_i)$. We refer to the projection as the local update. Note that $\widetilde{Z}$ does not impact the optimization of $q(\boldsymbol{f})$ or the GP hyper-parameters $\boldsymbol{\theta}$, so we omit the update formula for $\widetilde{Z}$. We summarize EP in algorithm 1 (Appendix). In section 4 we propose a new approximation approach which is similar to EP but employs the $L_2$ WD rather than the KL divergence.

### 3.3 Wasserstein Distance

We denote by $\mathcal{M}_+^1(\Omega)$ the set of all probability measures on $\Omega$. We consider probability measures on the $d$-dimensional real space $\mathbb{R}^d$. The WD between two probability distributions $\xi, \nu \in \mathcal{M}_+^1(\mathbb{R}^d)$ can be intuitively defined as the cost of transporting the probability mass from one distribution to the other. We are particularly interested in the subclass of $L_p$ WD, formally defined as follows.

**Definition 1** ($L_p$ WD). *Consider the set of all probability measures on the product space $\mathbb{R}^d \times \mathbb{R}^d$, whose marginal measures are $\xi$ and $\nu$ respectively, denoted as $U(\xi, \nu)$. The $L_p$ WD between $\xi$ and $\nu$ is defined as $W_p^p(\xi, \nu) \equiv \inf_{\pi \in U(\xi, \nu)} \int_{\mathbb{R}^d \times \mathbb{R}^d} \|\boldsymbol{x} - \boldsymbol{z}\|_p^p \, d\pi(\boldsymbol{x}, \boldsymbol{z})$ where $p \in [1, \infty)$ and $\|\cdot\|_p$ is the $L_p$ norm.*

Like the KL divergence, the $L_p$ WD it has a minimum of zero, achieved when the distributions are equivalent. *Unlike the KL*, however, it is a proper distance metric, and thereby satisfies the triangle inequality, and has the appealing property of symmetry.

A less fundamental property of the WD which we exploit for computational efficiency is:

**Proposition 1.** *[46, Remark 2.30] The $L_p$ WD between 1-d distribution functions $\xi$ and $\nu \in \mathcal{M}_+^1(\mathbb{R})$ equals the $L_p$ distance between the quantile functions, $W_p^p(\xi, \nu) = \int_0^1 \left| F_\xi^{-1}(y) - F_\nu^{-1}(y) \right|^p dy$, where $F_z : \mathbb{R} \to [0, 1]$ is the cumulative distribution function (CDF) of $z$, defined as $F_z(x) = \int_{-\infty}^x dz$, and $F_z^{-1}$ is the pseudoinverse or quantile function, defined as $F_z^{-1}(y) = \min_x\{x \in \mathbb{R} \cup \{-\infty\} : F_z(x) \geq y\}$.*

Finally, the following translation property of the $L_2$ WD is central to our proof of locality:

**Proposition 2.** *[46, Remark 2.19] Consider the $L_2$ WD defined for $\xi$ and $\nu \in \mathcal{M}_+^1(\mathbb{R}^d)$, and let $f_{\boldsymbol{\tau}}(\boldsymbol{x}) = \boldsymbol{x} - \boldsymbol{\tau}$, $\boldsymbol{\tau} \in \mathbb{R}^d$, be a translation operator. If $\xi_{\boldsymbol{\tau}}$ and $\nu_{\boldsymbol{\tau}'}$ denote the probability measures of translated random variables $f_{\boldsymbol{\tau}}(\boldsymbol{x})$, $\boldsymbol{x} \sim \xi$, and $f_{\boldsymbol{\tau}'}(\boldsymbol{x})$, $\boldsymbol{x} \sim \nu$, respectively, then $W_2^2(\xi_{\boldsymbol{\tau}}, \nu_{\boldsymbol{\tau}'}) = W_2^2(\xi, \nu) - 2(\boldsymbol{\tau} - \boldsymbol{\tau}')^{\mathsf{T}}(\boldsymbol{m}_\xi - \boldsymbol{m}_\nu) + \|\boldsymbol{\tau} - \boldsymbol{\tau}'\|_2^2$, where $\boldsymbol{m}_\xi$ and $\boldsymbol{m}_\nu$ are means of*

$\xi$ and $\nu$ respectively. In particular when $\boldsymbol{\tau} = \boldsymbol{m}_\xi$ and $\boldsymbol{\tau}' = \boldsymbol{m}_\nu$, $\xi_{\boldsymbol{\tau}}$ and $\nu_{\boldsymbol{\tau}'}$ become zero-mean measures, and $W_2^2(\xi_{\boldsymbol{\tau}}, \nu_{\boldsymbol{\tau}'}) = W_2^2(\xi, \nu) - \|\boldsymbol{m}_\xi - \boldsymbol{m}_\nu\|_2^2$.

# 4  Quantile Propagation

We now propose our new approximation algorithm which, as summarized in Algorithm 1 (Appendix), employs an $L_2$ WD based projection rather than the forward KL divergence projection of EP. Although QP employs a more complex divergence, it has the same computational complexity as EP, with the following caveat. To match the speed of EP, it is necessary to precompute sets of (data independent) lookup tables. Once precomputed, the resulting updates are only a constant factor slower than EP — a modest price to pay for optimizing a divergence which is challenging *even to evaluate*. Appendix J provides further details on the precomputation and use of these tables.

As stated in Proposition 1, minimizing $W_2^2(\widetilde{q}(f_i), \mathcal{N}(f_i))$ is equivalent to minimizing the $L_2$ distance between quantile functions of $\widetilde{q}(f_i)$ and $\mathcal{N}(f_i)$, so we refer to our method as quantile propagation (QP). This section focuses on deriving local updates for the site functions and analyzing their relationships with those of EP. Later in section 5, we show the locality property of QP, meaning that the site function $t(f)$ has a univariate parameterization and so the local update can be efficiently performed using marginals only.

## 4.1  Convexity of $L_p$ Wasserstein Distance

We first show $W_p^p(\widetilde{q}(f), \mathcal{N}(f|\mu, \sigma^2))$ to be strictly convex in $\mu$ and $\sigma$. Formally, we have:

**Theorem 1.** *Given two probability measures in $\mathcal{M}_+^1(\mathbb{R})$: a Gaussian $\mathcal{N}(\mu, \sigma^2)$ with mean $\mu$ and standard deviation $\sigma > 0$, and an arbitrary measure $\widetilde{q}$, $W_p^p(\widetilde{q}, \mathcal{N})$ is strictly convex in $\mu$ and $\sigma$.*

*Proof.*    See Appendix D.    □

## 4.2  Minimization of $L_2$ WD

An advantage of using the $L_p$ WD with $p = 2$, rather than other choices of $p$, is the computational efficiency it admits in the local updates. As we show in this section, optimizing the $L_2$ WD yields neat analytical updates of the optimal $\mu^\star$ and $\sigma^\star$ that require only univariate integration and the CDF of $\widetilde{q}(f)$. In contrast, other $L_p$ WDs lack convenient analytical expressions. Nonetheless, other $L_p$ WDs may have useful properties, the study of which we leave to future work.

The optimal parameters $\mu^\star$ and $\sigma^\star$ corresponding to the $L_2$ WD $W_2^2(\widetilde{q}, \mathcal{N}(\mu, \sigma^2))$ can be obtained using Proposition 1. Specifically, we employ the quantile function reformulation of $W_2^2(\widetilde{q}, \mathcal{N}(\mu, \sigma^2))$, and zero its derivatives w.r.t. $\mu$ and $\sigma$. The results provided below are derived in Appendix A:

$$\mu^\star = \mu_{\widetilde{q}} \;\; ; \;\; \sigma^\star = \sqrt{2} \int_0^1 F_{\widetilde{q}}^{-1}(y) \text{erf}^{-1}(2y - 1) \, \mathrm{d}y = 1/\sqrt{2\pi} \int_{-\infty}^{\infty} e^{-[\text{erf}^{-1}(2F_{\widetilde{q}}(f)-1)]^2} \, \mathrm{d}f. \quad (5)$$

Interestingly, the update for $\mu$ matches that of EP, namely the expectation under $\widetilde{q}$. However, for the standard deviation we have the difficulty of deriving the CDF $F_{\widetilde{q}}$. If a closed form expression is available, we can apply numerical integration to compute the optimal standard deviation; otherwise, we may use sampling based methods to approximate it. In our experiments we employ the former.

## 4.3  Properties of the Variance Update

Given the update equations in the previous section, here we show that the standard deviation estimate of QP, denoted as $\sigma_{\text{QP}}$, is less or equal to that of EP, denoted as $\sigma_{\text{EP}}$, when projecting the same tilted distribution to the Gaussian space.

**Theorem 2.** *The variances of the Gaussian approximation to a univariate tilted distribution $\widetilde{q}(f)$ as estimated by QP and EP satisfy $\sigma_{QP}^2 \le \sigma_{EP}^2$.*

*Proof.*    See Appendix E.    □

**Corollary 2.1.** *The variances of the site functions updated by EP and QP satisfy: $\widetilde{\sigma}_{QP}^2 \le \widetilde{\sigma}_{EP}^2$, and the variances of the approximate posterior marginals satisfy $\sigma_{q,QP}^2 \le \sigma_{q,EP}^2$.*

*Proof.* Since the cavity distribution is unchanged, we can calculate the variance of the site function as per Equation (4) and conclude that the variance of the site function also satisfies $\widetilde{\sigma}^2_{\mathrm{QP}} \leq \widetilde{\sigma}^2_{\mathrm{EP}}$. Moreover as per the definition of the cavity distribution in subsection 3.2, the approximate marginal distribution is proportional to the product of the cavity distribution and the site function $q(f_i) \propto q^{\backslash i}(f_i) t(f_i)$, which are two Gaussian distributions. By the product of Gaussians formula (Equation (4)), we know the variance of $q(f_i)$ estimated by EP as $\sigma^2_{q,\mathrm{EP}} = (\widetilde{\sigma}_{\mathrm{EP}}^{-2} + \sigma_{\backslash i}^{-2})^{-1} = \sigma^2_{\mathrm{EP}}$ and similarly $\sigma^2_{q,\mathrm{QP}} = \sigma^2_{\mathrm{QP}}$, where $\sigma^2_{\mathrm{EP}}$ and $\sigma^2_{\mathrm{QP}}$ are defined in Theorem E. Thus, there is $\sigma^2_{q,\mathrm{QP}} \leq \sigma^2_{q,\mathrm{EP}}$. $\square$

**Corollary 2.2.** *The predictive variances of latent functions at $\boldsymbol{x}_*$ by EP and QP satisfy:* $\sigma^2_{QP}(f(\boldsymbol{x}_*)) \leq \sigma^2_{EP}(f(\boldsymbol{x}_*))$.

*Proof.* The predictive variance of the latent function was analyzed in [47, Equation (3.61)]: $\sigma^2(f_*) = k_* - \boldsymbol{k}_*^\mathsf{T}(K + \widetilde{\Sigma})^{-1}\boldsymbol{k}_*$, where we define $f_* = f(\boldsymbol{x}_*)$ and $k_* = k(\boldsymbol{x}_*, \boldsymbol{x}_*)$, and let $\boldsymbol{k}_* = (k(\boldsymbol{x}_*, \boldsymbol{x}_i))_{i=1}^N$ be the (column) covariance vector between the test data $\boldsymbol{x}_*$ and the training data $\{\boldsymbol{x}_i\}_{i=1}^N$. After updating parameters of the site function $t_i(f_i)$, the predictive variance can be written as (details in Appendix I):

$$\sigma^2_{\mathrm{new}}(f_*) = k_* - \boldsymbol{k}_*^\mathsf{T} A \boldsymbol{k}_* + \boldsymbol{k}_*^\mathsf{T} \boldsymbol{s}_i \boldsymbol{s}_i^\mathsf{T} \boldsymbol{k}_* / [(\widetilde{\sigma}^2_{i,\mathrm{new}} - \widetilde{\sigma}^2_i)^{-1} + A_{ii}],$$

where $\widetilde{\sigma}^2_{i,\mathrm{new}}$ is the site variance updated by EP or QP, $A = (K + \widetilde{\Sigma})^{-1}$ and $\boldsymbol{s}_i$ is the $i$'s column of $A$. Since $\widetilde{\sigma}^2_{i,\mathrm{QP}} \leq \widetilde{\sigma}^2_{i,\mathrm{EP}}$, we have $\sigma^2_{\mathrm{QP}}(f_*) \leq \sigma^2_{\mathrm{EP}}(f_*)$. $\square$

**Remark.** *We compared variance estimates of EP and QP assuming the same cavity distribution. Proving analogous statements for the fixed points of the EP and QP algorithms is more challenging, however, and we leave this to future work, while providing empirical support for these analogous statements in Figure 1a. and Figure 1b.*

# 5 Locality Property

In this section we detail the central result on which our QP algorithm is based upon, which we refer to as the *locality property*. That is, the optimal site function $t_i$ is defined only in terms of the single corresponding latent variable $f_i$, and thereby and similarly to EP, it admits a simple and efficient sequential update of each individual site approximation.

## 5.1 Review: Locality Property of EP

We provide a brief review of the locality property of EP for GP models; for more details see Seeger [50]. We begin by defining the general site function $t_i(\boldsymbol{f})$ in terms of all of the latent variables, and the cavity and the tilted distributions as $q^{\backslash i}(\boldsymbol{f}) \propto p(\boldsymbol{f}) \prod_{j \neq i} \widetilde{t}_j(\boldsymbol{f})$ and $\widetilde{q}(\boldsymbol{f}) \propto q^{\backslash i}(\boldsymbol{f}) p(y_i | f_i)$, respectively. To update $t_i(\boldsymbol{f})$, EP matches a multivariate Gaussian distribution $\mathcal{N}(\boldsymbol{f})$ to $\widetilde{q}(\boldsymbol{f})$ by minimizing the KL divergence $\mathrm{KL}(\widetilde{q} \| \mathcal{N})$, which is further rewritten as (see details in Appendix F.1):

$$\mathrm{KL}\big(\widetilde{q} \| \mathcal{N}\big) = \mathrm{KL}\big(\widetilde{q}_i \| \mathcal{N}_i\big) + \mathbb{E}_{\widetilde{q}_i}\Big[\mathrm{KL}\big(q^{\backslash i}_{\backslash i | i} \| \mathcal{N}_{\backslash i | i}\big)\Big], \tag{6}$$

where and hereinafter, $\backslash i | i$ denotes the conditional distribution of $\boldsymbol{f}_{\backslash i}$ (taking $f_i$ out of $\boldsymbol{f}$) given $f_i$, namely, $q^{\backslash i}_{\backslash i | i} = q^{\backslash i}(\boldsymbol{f}_{\backslash i} | f_i)$ and $\mathcal{N}_{\backslash i | i} = \mathcal{N}(\boldsymbol{f}_{\backslash i} | f_i)$. Note that $q^{\backslash i}_{\backslash i | i}$ and $\mathcal{N}_{\backslash i | i}$ in the second term in Equation (6) are both Gaussian, and so setting them equal to one another causes that term to vanish. Furthermore, it is well known that the term $\mathrm{KL}\big(\widetilde{q}_i \| \mathcal{N}_i\big)$ is minimized w.r.t. the parameters of $\mathcal{N}_i$ by matching the first and second moments of $\widetilde{q}_i$ and $\mathcal{N}_i$. Finally, according to the usual EP logic, we recover the site function $t_i(\boldsymbol{f})$ by dividing the optimal Gaussian $\mathcal{N}(\boldsymbol{f})$ by the cavity $q^{\backslash i}(\boldsymbol{f})$:

$$t_i(\boldsymbol{f}) \propto \mathcal{N}(\boldsymbol{f})/q^{\backslash i}(\boldsymbol{f}) = \mathcal{N}(\boldsymbol{f}_{\backslash i}|f_i)\mathcal{N}(f_i)/(q^{\backslash i}(\boldsymbol{f}_{\backslash i}|f_i)q^{\backslash i}(f_i)) = \mathcal{N}(f_i)/q^{\backslash i}(f_i). \tag{7}$$

Here we can see the optimal site function $t_i(f_i)$ relies solely on the local latent variable $f_i$, so it is sufficient to assume a univariate expression for site functions. Besides, the site function can be efficiently updated by using the marginals $\widetilde{q}(f_i)$ and $\mathcal{N}(f_i)$ only, namely, $t_i(f_i) \propto \big(\min_{\mathcal{N}_i} \mathrm{KL}(\widetilde{q}_i \| \mathcal{N}_i)\big)/q^{\backslash i}(f_i)$.

## 5.2 Locality Property of QP

This section proves the locality property of QP, which turns out to be rather more involved to show than is the case for EP. We first prove the following theorem, and then follow the same procedure as for EP (Equation (7)).

**Theorem 3.** *Minimization of $W_2^2(\widetilde{q}(\boldsymbol{f}), \mathcal{N}(\boldsymbol{f}))$ w.r.t. $\mathcal{N}(\boldsymbol{f})$ results in $q^{\setminus i}(\boldsymbol{f}_{\setminus i}|f_i) = \mathcal{N}(\boldsymbol{f}_{\setminus i}|f_i)$.*

*Proof.* See Appendix F. $\qquad\square$

**Theorem 4** (Locality Property of QP). *For GP models with factorized likelihoods, QP requires only univariate site functions, and so yields efficient updates using only marginal distributions.*

*Proof.* We apply the same steps as in Equation (7) for the EP case to QP and we conclude that the site function $t_i(f_i) \propto \mathcal{N}(f_i)/q^{\setminus i}(f_i)$ relies solely on the local latent variable $f_i$. And as per Equation (22) (Appendix F), $\mathcal{N}(f_i)$ is estimated by $\min_{\mathcal{N}_i} W_2^2(\widetilde{q}_i, \mathcal{N}_i)$, so the local update only uses marginals and can perform efficiently. $\qquad\square$

**Benefits of the Locality Property.** The locality property admits an analytically economic form for the site function $t_i(f_i)$, requiring a parameterization that depends on a single latent variable. In addition, this also yields a significant reduction in the computational complexity, as only marginals are involved in each local update. In contrast, if QP (or EP) had no such a locality property, estimating the mean and the variance would involve integrals w.r.t. high-dimensional distributions, with a significantly higher computational cost should closed form expressions be unavailable.

# 6 Experiments

In this section, we compare the QP, EP and variational Bayes [VB, 42] algorithms for binary classification and Poisson regression. The experiments employ eight real world datasets and aim to compare relative accuracy of the three methods, rather than optimizing the absolute performance. The implementations of EP and VB in Python are publicly available [18], and our implementation of QP is based on that of EP. Our code is publicly available [1]. For both EP and QP, we stop local updates, *i.e.*, the inner loop in Algorithm 1 (Appendix), when the root mean squared change in parameters is less than $10^{-6}$. In the outer loop, the GP hyper-parameters are optimized by L-BFGS-B [6] with a maximum of $10^3$ iterations and a relative tolerance of $10^{-9}$ for the function value. VB is also optimized by L-BFGS-B with the same configuration. Parameters shared by the three methods are initialized to be the same.

## 6.1 Binary Classification

**Benchmark Data.** We perform binary classification experiments on the five real world datasets employed by Kuss and Rasmussen [28]: Ionosphere (IonoS), Wisconsin Breast Cancer, Sonar [13], Leptograpsus Crabs and Pima Indians Diabetes [48]. We use two additional UCI datasets as further evidence: Glass and Wine [13]. As the Wine dataset has three classes, we conduct binary classification experiments on all pairs of classes. We summarize the dataset size and data dimensions in Table 1.

**Prediction.** We predict the test labels using models optimized by EP, QP and VB on the training data. For a test input $\boldsymbol{x}_*$ with a binary target $y_*$, the approximate predictive distribution is written as: $q(y_*|\boldsymbol{x}_*) = \int_{-\infty}^{\infty} p(y_*|f_*)q(f_*)\,\mathrm{d}f_*$ where $f_* = f(\boldsymbol{x}_*)$ is the value of the latent function at $\boldsymbol{x}_*$. We use the probit likelihood for the binary classification task, which admits an analytical expression for the predictive distribution and results in a short-tailed posterior distribution. Correspondingly, the predicted label $\hat{y}_*$ is determined by thresholding the predictive probability at $1/2$.

**Performance Evaluation.** To evaluate the performance, we employ two measures: the test error (TE) and the negative test log-likelihood (NTLL). The TE and the NTLL quantify the prediction accuracy and uncertainty, respectively. Specifically, they are defined as $(\sum_{i=1}^{m} |y_{*,i} - \hat{y}_{*,i}|/2)/m$ and $-(\sum_{i=1}^{m} \log q(y_{*,i}|\boldsymbol{x}_{*,i}))/m$, respectively, for a set of test inputs $\{\boldsymbol{x}_{*,i}\}_{i=1}^{m}$, test labels $\{y_{*,i}\}_{i=1}^{m}$, and the predicted labels $\{\hat{y}_{*,i}\}_{i=1}^{m}$. Lower values indicate better performance for both measures.

Table 1: Results on benchmark datasets. The first three columns give dataset names, the number of instances $m$ and the number of features $n$. The table records the test errors (TEs) and the negative test log-likelihoods (NTLLs). The top section is on the benchmark datasets employed by Kuss and Rasmussen [28] and the middle section uses additional datasets. The bottom section shows Poisson regression results. * indicates that QP outperforms EP in more than 90% of experiments *consistently*.

| Data | m | n | TE ($\times 10^{-2}$) | | | NTLL($\times 10^{-3}$) | | |
|---|---|---|---|---|---|---|---|---|
| | | | EP | QP | VB | EP | QP | VB |
| IonoS | 351 | 34 | $\mathbf{7.9_{\pm 0.5}}$ | $\mathbf{7.9_{\pm 0.5}}$ | $18.9_{\pm 6.9}$ | $\mathbf{215.9_{\pm 8.4}}$ | $\mathbf{215.9_{\pm 8.5}}$ | $337.4_{\pm 70.8}$ |
| Cancer | 683 | 9 | $3.2_{\pm 0.2}$ | $3.2_{\pm 0.2}$ | $\mathbf{3.1_{\pm 0.2}}$ | $88.2_{\pm 3.1}$ | $\mathbf{88.2^{*}_{\pm 3.1}}$ | $88.9_{\pm 19.1}$ |
| Pima | 732 | 7 | $\mathbf{20.3_{\pm 1.0}}$ | $\mathbf{20.3_{\pm 1.0}}$ | $21.9_{\pm 0.4}$ | $424.7_{\pm 13.0}$ | $\mathbf{424.0^{*}_{\pm 13.2}}$ | $450.3_{\pm 2.6}$ |
| Crabs | 200 | 7 | $\mathbf{2.7_{\pm 0.5}}$ | $\mathbf{2.7_{\pm 0.5}}$ | $3.7_{\pm 0.7}$ | $64.4_{\pm 8.2}$ | $\mathbf{64.3_{\pm 8.4}}$ | $164.7_{\pm 7.5}$ |
| Sonar | 208 | 60 | $\mathbf{14.0_{\pm 1.1}}$ | $\mathbf{14.0_{\pm 1.1}}$ | $25.7_{\pm 3.9}$ | $306.7_{\pm 10.8}$ | $\mathbf{306.2^{*}_{\pm 10.9}}$ | $693.1_{\pm 0.0}$ |
| Glass | 214 | 10 | $1.1_{\pm 0.4}$ | $\mathbf{1.0_{\pm 0.4}}$ | $2.6_{\pm 0.5}$ | $29.5_{\pm 5.4}$ | $\mathbf{29.0^{*}_{\pm 5.5}}$ | $79.5_{\pm 6.3}$ |
| Wine1 | 130 | 13 | $\mathbf{1.5_{\pm 0.5}}$ | $\mathbf{1.5_{\pm 0.5}}$ | $1.7_{\pm 0.6}$ | $48.0_{\pm 3.4}$ | $\mathbf{47.4^{*}_{\pm 3.4}}$ | $83.9_{\pm 5.2}$ |
| Wine2 | 107 | 13 | $\mathbf{0.0_{\pm 0.0}}$ | $\mathbf{0.0_{\pm 0.0}}$ | $\mathbf{0.0_{\pm 0.0}}$ | $18.0_{\pm 1.2}$ | $\mathbf{17.8^{*}_{\pm 1.2}}$ | $26.7_{\pm 1.9}$ |
| Wine3 | 119 | 13 | $2.0_{\pm 1.0}$ | $2.0_{\pm 1.0}$ | $\mathbf{1.2_{\pm 0.7}}$ | $52.1_{\pm 5.6}$ | $\mathbf{51.8^{*}_{\pm 5.6}}$ | $69.4_{\pm 5.0}$ |
| Mining | 112 | 1 | $\mathbf{118.6_{\pm 27.0}}$ | $\mathbf{118.6_{\pm 27.0}}$ | $170.3_{\pm 15.9}$ | $1606.8_{\pm 116.3}$ | $\mathbf{1606.5_{\pm 116.3}}$ | $2007.3_{\pm 119.8}$ |

Note: `Wine1`: Class 1 vs. 2. `Wine2`: Class 1 vs. 3. `Wine3`: Class 2 vs. 3.

**Experiment Settings.** In the experiments, we randomly split each dataset into 10 folds, each time using 1 fold for testing and the other 9 folds for training, with features standardized to zero mean and unit standard deviation. We repeat this 100 times for a random seed ranging 0 through 99. As a result, there are a total of 1,000 experiments for each dataset. We report the average and the standard deviation of the above metrics over the 100 rounds.

**Results.** The evaluation results are summarized in Table 1. The top section presents the results on the datasets employed by Kuss and Rasmussen [28], whose reported TEs match ours as expected. While QP and EP exhibit similar TEs on these datasets, QP is superior to EP in terms of the NTLL. VB under-performs both EP and QP on all datasets except `Cancer`. The middle section of Table 1 shoes the results on additional datasets. The TEs are again similar for EP and QP, while QP has lower NTLLs. Again, VB performs worst among the three methods. To emphasize the difference between NTLLs of EP and QP, we mark with an asterisk those results in which QP outperforms EP in more than 90% of the experiments. Furthermore, we visualize the predictive variances of QP in comparison with those of EP in Figure 1a., which shows that the variances of QP are always less than or equal to those of EP, thereby providing empirical evidence of QP alleviating the over-estimation of predictive variances associated with the EP algorithm.

## 6.2 Poisson Regression

**Data and Settings.** We perform a Poisson regression experiment to further evaluate the performance of our method. The experiment employs the coal-mining disaster dataset [25] which has 190 data points indicating the time of fatal coal mining accidents in the United Kingdom from 1851 to 1962. To generate training and test sequences, we randomly assign every point of the original sequence to either a training or test sequence with equal probability, and this is repeated 200 times (random seeds $0, \cdots, 199$), resulting in 200 pairs of training and test sequences. We use the TE and the NTLL to evaluate the performance of the model on the test dataset. The NTLL has the same expression as that of the Binary classification experiment, but with a different predictive distribution $q(y_*|\boldsymbol{x}_*)$. The TE is defined slightly differently as $(\sum_{i=1}^{m} |y_{*,i} - \hat{y}_{*,i}|)/m$. To make the rate parameter of the Poisson likelihood non-negative, we use the square link function [15, 56], and as a result, the likelihood becomes $p(y|f^2)$. We use this link function because it is more mathematically convenient than the exponential function: the EP and QP update formulas, and the predictive distribution $q(y_*|\boldsymbol{x}_*)$ are available in Appendices C.2 and H, respectively.

**Results.** The means and the standard deviations of the evaluation results are reported in the last row of Table 1. Compared with EP, QP yields lower NTLL, which implies a better fitting performance of QP to the test sequences. We also provide the predictive variances in Figure 1b., in the variance of QP is once again seen to be less than or equal to that of EP. This experiment further supports our

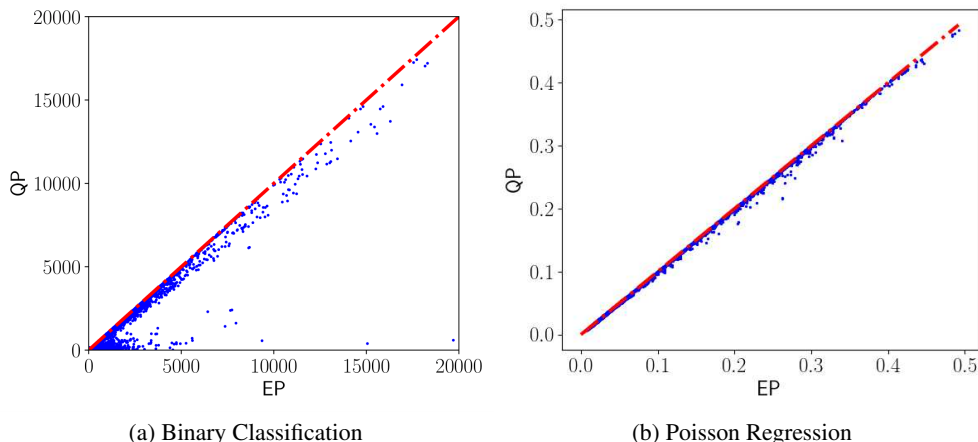

(a) Binary Classification          (b) Poisson Regression

Figure 1: A scatter plot of the predictive variances of latent functions on test data, for EP and QP. The diagonal dash line represents equivalence. We see that the predictive variance of QP is always less than or equal to that of EP.

claim that QP alleviates the problem with EP of over-estimation of the predictive variance. Finally, once again we find that both EP and QP outperform VB.

## 7 Conclusions

We have proposed QP as the first efficient $L_2$-WD based approximate Bayesian inference method for Gaussian process models with factorized likelihoods. Algorithmically, QP is similar to EP but uses the $L_2$ WD instead of the forward KL divergence for estimation of the site functions. When the likelihood factors are approximated by a Gaussian form we show that QP matches quantile functions rather than moments as in EP. Furthermore, we show that QP has the same mean update but a smaller variance than that of EP, which in turn alleviates the over-estimation by EP of the posterior variance in practice. Crucially, QP has the same favorable locality property as EP, and thereby admits efficient updates. Our experiments on binary classification and Poisson regression have shown that QP can outperform both EP and variational Bayes. Approximate inference with WD is promising but hard to compute, especially for continuous multivariate distributions. We believe our work paves the way for further practical approaches to WD-based inference.

**Limitations and Future Work**    Although we have presented properties and advantages of our method, it is still worth pointing out its limitations. First, our method does not provide a methodology for hyper-parameter optimization that is consistent with our proposed WD minimization framework. Instead, for this purpose, we rely on optimization of EP's marginal likelihood. We believe this is one of the reasons for the small performance differences between QP and EP.

Furthermore, the computational efficiency of our method comes at the price of additional memory requirements and the look-up tables may exhibit instabilities on high-dimensional data. To overcome these limitations, future work will explore alternatives to hyper-parameter optimization, improvements on numerical computation under the current approach and a variety of WD distances under a similar algorithm framework.

## Broader Impact

It is likely that the majority of significant technological advancements will eventually lead to both positive and negative societal and ethical outcomes. It is important, however, to consider how and when these outcomes may arise, and whether the net balance is likely to be favourable. After careful consideration, however, we found that the present work is sufficiently general and application independent, as to warrant relatively little specific concern.

## Acknowledgments

This research was undertaken with the assistance of resources from the National Computational Infrastructure (NCI Australia), an NCRIS enabled capability supported by the Australian Government. This work is also supported in part by ARC Discovery Project DP180101985 and Facebook Research under the Content Policy Research Initiative grants and conducted in partnership with the Defence Science and Technology Group, through the Next Generation Technologies Program.

## Footnotes

[1] `https://github.com/RuiZhang2016/Quantile-Propagation-for-Wasserstein-Approximate-Gaussian-Processes`

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
