[Supplementary Material · qp-si.pdf]

Supplements for *Quantile Propagation for Wasserstein-Approximate Gaussian Processes.*

## A  Minimization of $L_2$ WD between Univariate Gaussian and Non-Gaussian Distributions

In this section, we derive the formulas of the optimal $\mu^*$ and $\sigma^*$ for the $L_2$ WD, *i.e.*, Eqn. (5). Recall the optimization problem: we use a univariate Gaussian distribution $\mathcal{N}(f|\mu,\sigma^2)$ to approximate a univariate non-Gaussian distribution $q(f)$ by minimizing the $L_2$ WD between them:

$$\min_{\mu,\sigma} \mathrm{W}_2^2(q,\mathcal{N}) = \min_{\mu,\sigma} \int_0^1 \left| F_q^{-1}(y) - \mu - \sqrt{2}\sigma\mathrm{erf}^{-1}(2y-1) \right|^2 \mathrm{d}y,$$

where $F_q^{-1}$ is the quantile function of the non-Gaussian distribution $q$, namely the pseudoinverse function of the corresponding cumulative distribution function $F_q$ defined in Proposition 1.

To solve this problem, we first calculate derivatives about $\mu$ and $\sigma$:

$$\frac{\partial \mathrm{W}_2^2}{\partial \mu} = -2 \int_0^1 F_q^{-1}(y) - \mu - \sqrt{2}\sigma\mathrm{erf}^{-1}(2y-1) \, \mathrm{d}y,$$

$$\frac{\partial \mathrm{W}_2^2}{\partial \sigma} = -2 \int_0^1 (F_q^{-1}(y) - \mu - \sqrt{2}\sigma\mathrm{erf}^{-1}(2y-1))\sqrt{2}\mathrm{erf}^{-1}(2y-1) \, \mathrm{d}y.$$

Then, by zeroing derivatives, we obtain the optimal parameters:

$$\mu^* = \int_0^1 F_q^{-1}(y) - \sqrt{2}\sigma\mathrm{erf}^{-1}(2y-1) \, \mathrm{d}y$$

$$= \int_{-\infty}^{\infty} xq(x) \, \mathrm{d}x - \frac{\sqrt{2}}{2}\sigma \int_{-1}^{1} \mathrm{erf}^{-1}(y) \, \mathrm{d}y$$

$$= \mu_q - \sqrt{2}\sigma \int_{-\infty}^{\infty} x\mathcal{N}(x|0,1/2) \, \mathrm{d}x$$

$$= \mu_q,$$

$$\sigma^* = \sqrt{2} \int_0^1 (F_q^{-1}(y) - \mu)\mathrm{erf}^{-1}(2y-1) \, \mathrm{d}y \Big/ \int_0^1 2(\mathrm{erf}^{-1})^2(2y-1) \, \mathrm{d}y$$

$$= \sqrt{2} \int_0^1 F_q^{-1}(y)\mathrm{erf}^{-1}(2y-1) \, \mathrm{d}y \Big/ \underbrace{\int_{-\infty}^{\infty} 2x^2\mathcal{N}(x|0,1/2) \, \mathrm{d}x}_{=1}$$

$$= \sqrt{2} \int_0^1 F_q^{-1}(y)\mathrm{erf}^{-1}(2y-1) \, \mathrm{d}y$$

$$= \sqrt{2} \int_{-\infty}^{\infty} f\mathrm{erf}^{-1}(2F_q(f)-1) \, \mathrm{d}F_q(f)$$

$$= -\sqrt{\frac{1}{2\pi}} \int_{-\infty}^{\infty} f \, \mathrm{d}\, e^{-[\mathrm{erf}^{-1}(2F_{\tilde{q}}(f)-1)]^2}$$

$$= 0 + \sqrt{\frac{1}{2\pi}} \int_{-\infty}^{\infty} e^{-[\mathrm{erf}^{-1}(2F_{\tilde{q}}(f)-1)]^2} \, \mathrm{d}f. \tag{8}$$

## B  Minimization of $L_p$ WD between Univariate Gaussian and Non-Gaussian Distributions

In this section, we describe a gradient descent approach to minimizing an $L_p$ WD, for $p \neq 2$, in order to handle cases with no analytical expressions for the optimal parameters. Our goal is to use a univariate Gaussian distribution $\mathcal{N}(f|\mu,\sigma^2)$ to approximate a univariate non-Gaussian distribution $q(f)$. Specifically, we seek the minimiser in $\mu$ and $\sigma$ of $\mathrm{W}_p^p(q,\mathcal{N})$; the derivatives of the objective

function about $\mu$ and $\sigma$ are:

$$\partial_\mu W_p^p = -p \int_0^1 |\varepsilon(y)|^{p-1}\mathrm{sgn}(\varepsilon(y)) \, \mathrm{d}y = -p \int_{-\infty}^\infty |\eta(x)|^{p-1}\mathrm{sgn}(\eta(x))q(x) \, \mathrm{d}x,$$

$$\partial_\sigma W_p^p = -p \int_0^1 |\varepsilon(y)|^{p-1}\mathrm{sgn}(\varepsilon(y))\mathrm{erf}^{-1}(2y-1) \, \mathrm{d}y = -p \int_{-\infty}^\infty |\eta(x)|^{p-1}\mathrm{sgn}(\eta(x))\mathrm{erf}^{-1}(2F_q(x)-1)q(x) \, \mathrm{d}x.$$

where for simplification, we define $\varepsilon(y) = F_q^{-1}(y) - \mu - \sqrt{2}\sigma\mathrm{erf}^{-1}(2y-1)$ and $\eta(x) = x - \mu - \sqrt{2}\sigma\mathrm{erf}^{-1}(2F_q(x)-1)$, with $F_q$ and $F_q^{-1}$ being the CDF and the quantile function of $q$. Note the derivatives have no analytical expressions. However, if the CDF $F_q$ is available, we can use the standard numerical integration routines; otherwise, we resort to Monte Carlo sampling. In the framework of EP or QP, $q(x) \propto q^{\backslash i}(x)p(y_i|x)$ and $q^{\backslash i}$ is Gaussian, so we may draw samples from a Gaussian proposal distribution to obtain a simple Monte Carlo method.

## C  Computations for Different Likelihoods

Given the likelihood $p(y|f)$ and the cavity distribution $q^{\backslash i}(f) = \mathcal{N}(f|\mu,\sigma^2)$, a stable way to compute the mean and the variance of the tilted distribution $\widetilde{q}(f) = p(y|f)q^{\backslash i}(f)/Z$ where the normalizer $Z = \int_{-\infty}^\infty p(y|f)q^{\backslash i}(f) \, \mathrm{d}f$, can be found in the software manual of Rasmussen and Williams [47]. We present the key formulae below, for use in subsequent derivations:

$$\partial_\mu Z = \int_{-\infty}^\infty \frac{f-\mu}{\sigma^2}p(y|f)\mathcal{N}(f|\mu,\sigma^2) \, \mathrm{d}f$$

$$\frac{\partial_\mu Z}{Z} = \frac{1}{\sigma^2}\int_{-\infty}^\infty f\frac{p(y|f)\mathcal{N}(f|\mu,\sigma^2)}{Z} \, \mathrm{d}f - \frac{\mu}{\sigma^2}\int_{-\infty}^\infty \frac{p(y|f)\mathcal{N}(f|\mu,\sigma^2)}{Z} \, \mathrm{d}y$$

$$\frac{\partial_\mu Z}{Z} = \frac{1}{\sigma^2}\mu_{\widetilde{q}} - \frac{\mu}{\sigma^2}$$

$$\implies \mu_{\widetilde{q}} = \frac{\sigma^2\partial_\mu Z}{Z} + \mu = \sigma^2\partial_\mu \log Z + \mu,$$

$$\partial_\mu^2 Z = \int_{-\infty}^\infty -\frac{1}{\sigma^2}p(y|f)\mathcal{N}(f|\mu,\sigma^2) + \left(\frac{f-\mu}{\sigma^2}\right)^2 p(y|f)\mathcal{N}(f|\mu,\sigma^2) \, \mathrm{d}f$$

$$\frac{\partial_\mu^2 Z}{Z} = \int_{-\infty}^\infty \left(-\frac{1}{\sigma^2} + \frac{\mu^2}{\sigma^4} + \frac{f^2}{\sigma^4} - \frac{2\mu f}{\sigma^4}\right)\frac{p(y|f)\mathcal{N}(f|\mu,\sigma^2)}{Z} \, \mathrm{d}f$$

$$\frac{\partial_\mu^2 Z}{Z} = -\frac{1}{\sigma^2} + \frac{\mu^2}{\sigma^4} + \frac{1}{\sigma^4}(\sigma_{\widetilde{q}}^2 + \mu_{\widetilde{q}}^2) - \frac{2\mu}{\sigma^4}\mu_{\widetilde{q}}$$

$$\frac{\partial_\mu^2 Z}{Z} = -\frac{1}{\sigma^2} + \frac{\sigma_{\widetilde{q}}^2}{\sigma^4} + \frac{(\mu-\mu_{\widetilde{q}})^2}{\sigma^4} = -\frac{1}{\sigma^2} + \frac{\sigma_{\widetilde{q}}^2}{\sigma^4} + \left(\frac{\partial_\mu Z}{Z}\right)^2$$

$$\implies \sigma_{\widetilde{q}}^2 = \sigma^4\left[\frac{\partial_\mu^2 Z}{Z} - \left(\frac{\partial_\mu Z}{Z}\right)^2\right] + \sigma^2 = \sigma^4\partial_\mu^2 \log Z + \sigma^2.$$

### C.1  Probit Likelihood for Binary Classification

For the binary classification with labels $y \in \{-1, 1\}$, the PDF of the tilted distribution $\widetilde{q}(f)$ with the probit likelihood is provided by Rasmussen and Williams [47]:

$$\widetilde{q}(f) = Z^{-1}\Phi(fy)\mathcal{N}(f|\mu,\sigma^2), \quad Z = \Phi(z), \quad z = \frac{\mu}{y\sqrt{1+\sigma^2}},$$

and the mean estimate also has a closed form expression:

$$\mu^\star = \mu_{\widetilde{q}} = \mu + \frac{\sigma^2\mathcal{N}(z)}{\Phi(z)y\sqrt{1+\sigma^2}}.$$

As per Equation (5), the computation of the optimal $\sigma^\star$ requires the CDF of $\widetilde{q}$, denoted as $F_{\widetilde{q}}$. For positive $y > 0$, the CDF is derived as

$$F_{\widetilde{q}, y>0}(x) = Z^{-1} \int_{-\infty}^{x} \Phi(fy) \, \mathcal{N}\left(f|\mu, \sigma^2\right) \, \mathrm{d}f$$

$$= \frac{Z^{-1}}{2\pi\sigma y} \int_{-\infty}^{\mu} \int_{-\infty}^{x-\mu} \exp\left(-\frac{1}{2} \begin{bmatrix} w \\ f \end{bmatrix}^T \begin{bmatrix} v^{-2} + \sigma^{-2} & v^{-2} \\ v^{-2} & v^{-2} \end{bmatrix} \begin{bmatrix} w \\ f \end{bmatrix}\right) \, \mathrm{d}w \, \mathrm{d}f$$

$$= Z^{-1} \int_{-\infty}^{k} \int_{-\infty}^{h} \mathcal{N}\left(\begin{bmatrix} w \\ f \end{bmatrix} \middle| \mathbf{0}, \begin{bmatrix} 1 & -\rho \\ -\rho & 1 \end{bmatrix}\right) \, \mathrm{d}w \, \mathrm{d}f$$

$$\overset{(a)}{=} Z^{-1} \left[\frac{1}{2}\Phi(h) - T\left(h, \frac{k+\rho h}{h\sqrt{1-\rho^2}}\right) + \frac{1}{2}\Phi(k) - T\left(k, \frac{h+\rho k}{k\sqrt{1-\rho^2}}\right) + \eta\right]$$

$$k = \frac{\mu}{\sqrt{\sigma^2+1}}, \quad h = \frac{x-\mu}{\sigma}, \quad \rho = \frac{1}{\sqrt{1+1/\sigma^2}}, \quad x \neq \mu, \ \mu \neq 0,$$

where the step (a) is obtained by exploiting the work of Owen [45] and $T(\cdot, \cdot)$ is the Owen's T function:

$$T(h, a) = \frac{1}{2\pi} \int_0^a \frac{\exp\left[-(1+x^2)h^2/2\right]}{1+x^2} \, \mathrm{d}x,$$

and $\eta$ is defined as

$$\eta = \begin{cases} 0 & hk > 0 \text{ or } (hk = 0 \text{ and } h+k \geq 0), \\ -0.5 & \text{otherwise.} \end{cases}$$

Similarly, for $y < 0$, the CDF is

$$F_{\widetilde{q}, y<0}(x) = Z^{-1} \left[\frac{1}{2}\Phi(h) + T\left(h, \frac{k+\rho h}{h\sqrt{1-\rho^2}}\right) - \frac{1}{2}\Phi(k) + T\left(k, \frac{h+\rho k}{k\sqrt{1-\rho^2}}\right) - \eta\right].$$

Summarizing the two cases, we get the closed form expression of $F_{\widetilde{q}}$:

$$F_{\widetilde{q}}(x) = Z^{-1} \left[\frac{1}{2}\Phi(h) - yT\left(h, \frac{k+\rho h}{h\sqrt{1-\rho^2}}\right) + \frac{y}{2}\Phi(k) - yT\left(k, \frac{h+\rho k}{k\sqrt{1-\rho^2}}\right) + y\eta\right]$$

$$= Z^{-1} \left[\frac{1}{2}\Phi(h) - yT\left(h, \frac{k}{h\sqrt{1-\rho^2}} + \sigma\right) + \frac{y}{2}\Phi(k) - yT\left(k, \frac{h}{k\sqrt{1-\rho^2}} + \sigma\right) + y\eta\right].$$

Provided the above, the optimal $\sigma^\star$ can be computed by numerical integration of Eqn (8). For special cases, we provide additional formulas:

$$(1) \, x = \mu, \ \mu \neq 0 : F_{\widetilde{q}}(x) = Z^{-1} \left[\frac{1}{4} - \frac{y\mathrm{sign}(k)}{4} + \frac{y}{2}\Phi(k) - yT(k, \sigma) + y\eta\right];$$

$$(2) \, x \neq \mu, \ \mu = 0 : F_{\widetilde{q}}(x) = 2 \left[\frac{1}{2}\Phi(h) - yT(h, \sigma) + \frac{y}{4} - \frac{y\mathrm{sign}(h)}{4} + y\eta\right];$$

$$(3) \, x = \mu, \ \mu = 0 : F_{\widetilde{q}}(x) = \frac{1}{2} - \frac{y}{\pi}\arctan(\sigma).$$

## C.2 Square Link Function for Poisson Regression

Consider Poisson regression, which uses the Poisson likelihood $p(y|g) = g^y \exp(-g)/y!$ to model count data $y \in \mathbb{N}$, with the square link function $g(f) = f^2$ [56, 15]. We use the square link function because it is more mathematically convenient than the exponential function. Given the cavity distribution $q^{\backslash i}(f) = \mathcal{N}(f|\mu, \sigma^2)$, we want the tilted distribution $\widetilde{q}(f) = q^{\backslash i}(f)p(y|g(f))/Z$ where the normalizer $Z$ is derived as:

$$Z = \int_{-\infty}^{\infty} q^{\backslash i}(f)p(y|g) \, \mathrm{d}f$$

$$= \int_{-\infty}^{\infty} \frac{1}{\sqrt{2\pi\sigma^2}} \exp\left(-\frac{(f-\mu)^2}{2\sigma^2}\right) f^{2y} \exp(-f^2)/y! \, \mathrm{d}f$$

$$\overset{(a)}{=} \frac{1}{\sqrt{2\pi\sigma^2} y! \exp(\mu^2/(1+2\sigma^2))} \int_{-\infty}^{\infty} f^{2y} \exp\left(-\frac{(f-\mu/(1+2\sigma^2))^2}{2\sigma^2/(1+2\sigma^2)}\right) \mathrm{d}f$$

$$\overset{(b)}{=} \frac{\left(\frac{2\sigma^2}{1+2\sigma^2}\right)^{y+\frac{1}{2}}}{\sqrt{2\pi\sigma^2} y! \exp(\mu^2/(1+2\sigma^2))} \Gamma\left(y+\frac{1}{2}\right) {}_1F_1\left(-y; \frac{1}{2}; -\frac{\mu^2}{2\sigma^2(1+2\sigma^2)}\right)$$

$$= \frac{\alpha^{y+\frac{1}{2}}}{\sqrt{2\pi\sigma^2} y! \exp(h)} \Gamma\left(y+\frac{1}{2}\right) {}_1F_1\left(-y; \frac{1}{2}; -\frac{h}{2\sigma^2}\right),$$

$$\alpha = \frac{2\sigma^2}{1+2\sigma^2}, \quad h = \frac{\mu^2}{1+2\sigma^2} \tag{9}$$

where the step (a) rewrites the product of two exponential functions into the form of the Gaussian distribution, (b) is achieved through Mathematica [59], $\Gamma(\cdot)$ is the Gamma function and ${}_1F_1\left(-y; \frac{1}{2}; -\frac{h^2}{2\sigma^2}\right)$ is the confluent hypergeometric function of the first kind. Furthermore, we compute the first derivative of $\log Z$ w.r.t. $\mu$ and then the mean of the tilted distribution:

$$\partial_\mu \log Z = \left(\frac{y\, {}_1F_1\left(-y+1; \frac{3}{2}; -\frac{h}{2\sigma^2}\right)}{\sigma^2\, {}_1F_1\left(-y; \frac{1}{2}; -\frac{h}{2\sigma^2}\right)} - 1\right) \frac{2\mu}{1+2\sigma^2}$$

$$\implies \mu_{\widetilde{q}} = \sigma^2 \partial_\mu \log Z + \mu.$$

$$\partial_\mu^2 \log Z = \left(\frac{y\, {}_1F_1\left(-y+1; \frac{3}{2}; -\frac{h}{2\sigma^2}\right)}{\sigma^2\, {}_1F_1\left(-y; \frac{1}{2}; -\frac{h}{2\sigma^2}\right)} - 1\right) \frac{2}{1+2\sigma^2} - $$

$$\left(\frac{2(1-y)\, {}_1F_1\left(-y+2; \frac{5}{2}; -\frac{h}{2\sigma^2}\right)}{3\, {}_1F_1\left(-y; \frac{1}{2}; -\frac{h}{2\sigma^2}\right)} + \frac{2y\, {}_1F_1\left(-y+1; \frac{3}{2}; -\frac{h}{2\sigma^2}\right)^2}{{}_1F_1\left(-y; \frac{1}{2}; -\frac{h}{2\sigma^2}\right)^2}\right) \frac{2\mu^2 y}{\sigma^4 (1+2\sigma^2)^2}$$

$$\implies \sigma_{\widetilde{q}}^2 = \sigma^4 \partial_\mu^2 \log Z + \sigma^2$$

Finally, we derive the CDF of the tilted distribution $\widetilde{q}$ by using the binomial theorem:

$$F_{\widetilde{q}}(x) = Z^{-1} \int_{-\infty}^{x} p(y|g) \mathcal{N}(f|\mu, \sigma^2) \, \mathrm{d}f$$

$$\overset{(a)}{=} A \int_{-\infty}^{x} f^{2y} \exp\left(-\frac{(f-\mu/(1+2\sigma^2))^2}{2\sigma^2/(1+2\sigma^2)}\right) \mathrm{d}f$$

$$= A \int_{-\infty}^{x-\frac{\mu}{1+2\sigma^2}} \left(f + \frac{\mu}{1+2\sigma^2}\right)^{2y} \exp\left(-\frac{f^2}{2\sigma^2/(1+2\sigma^2)}\right) \mathrm{d}f$$

$$\overset{(b)}{=} A \int_{-\infty}^{x-\beta} \left[\sum_{k=0}^{2y} \binom{2y}{k} f^k \beta^{2y-k}\right] \exp\left(-\frac{f^2}{\alpha}\right) \mathrm{d}f$$

$$= A \sum_{k=0}^{2y} \binom{2y}{k} \beta^{2y-k} \left[\int_{-\infty}^{0} f^k \exp\left(-\frac{f^2}{\alpha}\right) \mathrm{d}f + \int_{0}^{x-\beta} f^k \exp\left(-\frac{f^2}{\alpha}\right) \mathrm{d}f\right]$$

$$\overset{(c)}{=} \frac{A}{2} \sum_{k=0}^{2y} \binom{2y}{k} \beta^{2y-k} \alpha^{\frac{k+1}{2}} \left[(-1)^k \Gamma\left(\frac{k+1}{2}\right) + \mathrm{sgn}(x-\beta)^{k+1} \left(\Gamma\left(\frac{k+1}{2}\right) - \Gamma\left(\frac{k+1}{2}, \frac{(x-\beta)^2}{\alpha}\right)\right)\right]$$

$$A = \frac{Z^{-1}}{\sqrt{2\pi\sigma^2} y! \exp(\mu^2/(1+2\sigma^2))} = \left[\alpha^{y+\frac{1}{2}} \Gamma\left(y+\frac{1}{2}\right) {}_1F_1\left(-y; \frac{1}{2}; -\frac{h}{2\sigma^2}\right)\right]^{-1}, \quad \beta = \frac{\mu}{1+2\sigma^2},$$

where the step (a) has been derived in (a) of Eqn. (9), (b) applies the binomial theorem and (c) is obtained through Mathematica [59]. And, the function $\Gamma(a, z) = \int_{z}^{\infty} t^{a-1} e^{-t} \, \mathrm{d}t$ is the upper

incomplete gamma function and $\text{sgn}(x)$ is the sign function, equaling 1 when $x > 0$, 0 when $x = 0$ and $-1$ when $x < 0$.

## D   Proof of Convexity

**Theorem**   Given two probability measures in $\mathcal{M}_+^1(\mathbb{R})$: a Gaussian $\mathcal{N}(\mu, \sigma^2)$ with mean $\mu$ and standard deviation $\sigma > 0$, and an arbitrary measure $\widetilde{q}$, the $L_p$ WD $\text{W}_p^p(\widetilde{q}, \mathcal{N})$ is strictly convex about $\mu$ and $\sigma$.

*Proof.*   Let $F_{\widetilde{q}}^{-1}(y)$ and $F_{\mathcal{N}}^{-1}(y) = \mu + \sqrt{2}\sigma\text{erf}^{-1}(2y - 1)$, $y \in [0, 1]$, be the quantile functions of $\widetilde{q}$ and the Gaussian $\mathcal{N}$, where erf is the error function. Then, we consider two distinct Gaussian measures $\mathcal{N}(\mu_1, \sigma_1^2)$ and $\mathcal{N}(\mu_2, \sigma_2^2)$ and a convex combination w.r.t. their parameters $\mathcal{N}(a_1\mu_1 + a_2\mu_2, (a_1\sigma_1 + a_2\sigma_2)^2)$ with $a_1, a_2 \in \mathbb{R}_+$ and $a_1 + a_2 = 1$. Given the above, we further define $\varepsilon_k(y) = F_{\widetilde{q}}^{-1}(y) - \mu_k - \sigma_k\sqrt{2}\text{erf}^{-1}(2y - 1)$, $k = 1, 2$, for notational simplification, and derive the convexity as:

$$\text{W}_p^p(\widetilde{q}, \mathcal{N}(a_1\mu_1 + a_2\mu_2, (a_1\sigma_1 + a_2\sigma_2)^2)) \stackrel{(a)}{=} \int_0^1 |a_1\varepsilon_1(y) + a_2\varepsilon_2(y)|^p \, \mathrm{d}y \stackrel{(b)}{\le} \int_0^1 (a_1|\varepsilon_1(y)| +$$

$$a_2|\varepsilon_2(y)|)^p \, \mathrm{d}y \stackrel{(c)}{\le} a_1\text{W}_p^p(\widetilde{q}, \mathcal{N}(\mu_1, \sigma_1^2)) + a_2\text{W}_p^p(\widetilde{q}, \mathcal{N}(\mu_2, \sigma_2^2)),$$

where steps (a), (b) and (c) are obtained by applying Proposition 1, non-negativity of the absolute value, and the convexity of $f(x) = x^p$, $p \ge 1$, over $\mathbb{R}_+$ respectively. The equality at $(b)$ holds iff $\varepsilon_k(y) \ge 0, k = 1, 2, \forall y \in [0, 1]$, and $(c)$'s equality holds iff $|\varepsilon_1(y)| = |\varepsilon_2(y)|$, $\forall y \in [0, 1]$. These two conditions for equality can't be attained simultaneously as otherwise it would contradict that $\mathcal{N}(\mu_1, \sigma_1^2)$ is different from $\mathcal{N}(\mu_2, \sigma_2^2)$. So, $\text{W}_p^p(\widetilde{q}, \mathcal{N})$, $p \ge 1$, is strictly convex about $\mu$ and $\sigma$.   $\square$

## E   Proof of Variance Difference

**Theorem**   The variance of the Gaussian approximation to a univariate tilted distribution $\widetilde{q}(f)$ as estimated by QP and EP satisfy $\sigma_{\text{QP}}^2 \le \sigma_{\text{EP}}^2$.

*Proof.*   Let $\mathcal{N}(\mu_{\text{QP}}, \sigma_{\text{QP}}^2)$ be the optimal Gaussian in QP. As per Proposition 1, we reformulate the $L_2$ WD based projection $\text{W}_2^2(\widetilde{q}, \mathcal{N}(\mu_{\text{QP}}, \sigma_{\text{QP}}^2))$ w.r.t. quantile functions:

$$\text{W}_2^2(\widetilde{q}, \mathcal{N}(\mu_{\text{QP}}, \sigma_{\text{QP}}^2)) = \int_0^1 |F_{\widetilde{q}}^{-1}(y) - \mu_{\text{QP}} - \sqrt{2}\sigma_{\text{QP}}\text{erf}^{-1}(2y-1)|^2 \, \mathrm{d}y = \int_0^1 \underbrace{(F_{\widetilde{q}}^{-1}(y) - \mu_{\text{QP}})^2}_{\sigma_{\text{EP}}^2}$$

$$+ \underbrace{(\sqrt{2}\sigma_{\text{QP}}\text{erf}^{-1}(2y-1))^2}_{\sigma_{\text{QP}}^2} - \underbrace{2(F_{\widetilde{q}}^{-1}(y) - \mu_{\text{QP}})\sqrt{2}\sigma_{\text{QP}}\text{erf}^{-1}(2y-1)}_{(\text{A})} \, \mathrm{d}y = \sigma_{\text{EP}}^2 - \sigma_{\text{QP}}^2,$$

where for (A), we used $\int \mu_{\text{QP}}\sigma_{\text{QP}}\text{erf}^{-1}(2y - 1) \, \mathrm{d}y = 0$ and the remaining factor can be easily shown to be equal to $2\sigma_{\text{QP}}^2$. Furthermore, due to the non-negativity of the WD, we have $\sigma_{\text{EP}}^2 \ge \sigma_{\text{QP}}^2$, and the equality holds iff $\widetilde{q}$ is Gaussian.   $\square$

## F   Proof of Locality Property

**Theorem**   Minimization of $\text{W}_2^2(\widetilde{q}(\boldsymbol{f}), \mathcal{N}(\boldsymbol{f}))$ w.r.t. $\mathcal{N}(\boldsymbol{f})$ results in $q^{\backslash i}(\boldsymbol{f}_{\backslash i}|f_i) = \mathcal{N}(\boldsymbol{f}_{\backslash i}|f_i)$.

*Proof.*   We first apply the decomposition of the $L_2$ norm to rewriting the $\text{W}_2^2(\widetilde{q}(\boldsymbol{f}), \mathcal{N}(\boldsymbol{f}))$ as below (see detailed derivations in Appendix F.2):

$$\text{W}_2^2(\widetilde{q}, \mathcal{N}) = \inf_{\pi_i} \mathbb{E}_{\pi_i}\left[\|f_i - f_i'\|_2^2 + \text{W}_2^2(q_{\backslash i|i}^{\backslash i}, \mathcal{N}_{\backslash i|i})\right], \tag{10}$$

where the prime indicates that the variable is from the Gaussian $\mathcal{N}$, and for simplification, we use the notation $\pi_i$ for the joint distribution $\pi(f_i, f_i')$ which belongs to a set of measures $U(\widetilde{q}_i, \mathcal{N}_i)$. Since

$q^{\backslash i}(\boldsymbol{f})$ is known to be Gaussian, we define it in a partitioned form:

$$q^{\backslash i}(\boldsymbol{f}) \equiv \mathcal{N}\left(\begin{bmatrix} \boldsymbol{f}_{\backslash i} \\ f_i \end{bmatrix} \middle| \begin{bmatrix} \boldsymbol{m}_{\backslash i} \\ m_i \end{bmatrix}, \begin{bmatrix} \boldsymbol{S}_{\backslash i} & \boldsymbol{S}_{\backslash ii} \\ \boldsymbol{S}_{\backslash ii}^{\mathsf{T}} & S_i \end{bmatrix}\right), \tag{11}$$

and the conditional $q^{\backslash i}(\boldsymbol{f}_{\backslash i}|f_i)$ is expressed as:

$$q^{\backslash i}(\boldsymbol{f}_{\backslash i}|f_i) = \mathcal{N}(\boldsymbol{f}_{\backslash i}|\boldsymbol{m}_{\backslash i|i}, \boldsymbol{S}_{\backslash i|i}), \quad \boldsymbol{m}_{\backslash i|i} = \boldsymbol{m}_{\backslash i} + \boldsymbol{S}_{\backslash ii}S_i^{-1}(f_i - m_i) \equiv \boldsymbol{a}f_i + \boldsymbol{b}, \tag{12}$$

$$\boldsymbol{S}_{\backslash i|i} = \boldsymbol{S}_{\backslash i} - \boldsymbol{S}_{\backslash ii}S_i^{-1}\boldsymbol{S}_{\backslash ii}^{\mathsf{T}}.$$

We define a similar partitioned expression for the Gaussian $\mathcal{N}(\boldsymbol{f}')$ by adding primes to variables and parameters on the r.h.s. of Equation (11), and as a result, the conditional $\mathcal{N}(\boldsymbol{f}'_{\backslash i}|f'_i)$ is written as:

$$\mathcal{N}(\boldsymbol{f}'_{\backslash i}|f'_i) = \mathcal{N}(\boldsymbol{m}'_{\backslash i|i}, \boldsymbol{S}'_{\backslash i|i}), \quad \boldsymbol{m}'_{\backslash i|i} = \boldsymbol{m}'_{\backslash i} + \boldsymbol{S}'_{\backslash ii}S_i'^{-1}(f'_i - m'_i) \equiv \boldsymbol{a}'f'_i + \boldsymbol{b}', \tag{13}$$

$$\boldsymbol{S}'_{\backslash i|i} = \boldsymbol{S}'_{\backslash i} - \boldsymbol{S}'_{\backslash ii}S_i'^{-1}\boldsymbol{S}'^{\mathsf{T}}_{\backslash ii}. \tag{14}$$

Given the above definitions, we exploit Proposition 2 to take the means out of the $L_2$ WD on the r.h.s. of Equation (10):

$$\mathrm{W}_2^2(\widetilde{q}, \mathcal{N}) = \inf_{\pi_i} \mathbb{E}_{\pi_i}\left[\|f_i - f'_i\|_2^2 + \|\boldsymbol{m}_{\backslash i|i} - \boldsymbol{m}'_{\backslash i|i}\|_2^2\right] + \underbrace{\mathrm{W}_2^2\left(\mathcal{N}(\boldsymbol{0}, \boldsymbol{S}_{\backslash i|i}), \mathcal{N}(\boldsymbol{0}, \boldsymbol{S}'_{\backslash i|i})\right)}_{(A)}. \tag{15}$$

Minimizing this function requires optimizing $m'_i$, $\boldsymbol{m}'_{\backslash i}$, $S'_i$, $\boldsymbol{S}'_{\backslash i}$ and $\boldsymbol{S}'_{\backslash ii}$. As $\boldsymbol{S}'_{\backslash i}$ is only contained in $\boldsymbol{S}_{\backslash i|i}$ and isolated into the term $(A)$, it can be optimized by simply setting

$$\boldsymbol{S}'_{\backslash i|i} = \boldsymbol{S}_{\backslash i|i} \overset{\text{Eqn. (14)}}{\Longrightarrow} \boldsymbol{S}^{(n)*}_{\backslash i} = \boldsymbol{S}_{\backslash i|i} + \boldsymbol{S}'_{\backslash ii}S_i'^{-1}\boldsymbol{S}'^{\mathsf{T}}_{\backslash ii}. \tag{16}$$

As a result, $(A)$ is minimized to zero. Next, we plug in expressions of $\boldsymbol{m}_{\backslash i|i}$ and $\boldsymbol{m}'_{\backslash i|i}$ (Equation (12) and Equation (13)) into optimized Equation (15):

$$\min_{\boldsymbol{S}'_{\backslash i}}(15) = \inf_{\pi_i}\mathbb{E}_{\pi_i}\left[\|f_i - f'_i\|_2^2 + \|\boldsymbol{a}f_i - \boldsymbol{a}'f'_i + \boldsymbol{b} - \boldsymbol{b}'\|_2^2\right], \tag{17}$$

where $\boldsymbol{m}'_{\backslash i}$ is only contained by $\boldsymbol{b}'$. Thus, we can optimize it by zeroing the derivative of the above function about $\boldsymbol{m}'_{\backslash i}$, which results in:

$$\boldsymbol{b}' = \boldsymbol{b} + \boldsymbol{a}\mu_{\widetilde{q}_i} - \boldsymbol{a}'m'_i \overset{\text{Eqn. (13)}}{\Longrightarrow} \boldsymbol{m}^{(n)*}_{\backslash i} = \boldsymbol{S}'_{\backslash ii}S_i'^{-1}m'_i + \boldsymbol{b} + \boldsymbol{a}\mu_{\widetilde{q}_i} - \boldsymbol{a}'m'_i, \tag{18}$$

where $\mu_{\widetilde{q}_i}$ is the mean of $\widetilde{q}(f_i)$. The minimum value of Equation (17) thereby is (see details in subsection F.3):

$$\min_{\boldsymbol{m}'_{\backslash i}}(17) = (1 + \boldsymbol{a}^{\mathsf{T}}\boldsymbol{a}')\mathrm{W}_2^2(\widetilde{q}_i, \mathcal{N}_i) + \|\boldsymbol{a}\|_2^2\sigma_{\widetilde{q}_i}^2 + \|\boldsymbol{a}'\|_2^2 S'_i - \boldsymbol{a}^{\mathsf{T}}\boldsymbol{a}'\left[\sigma_{\widetilde{q}_i}^2 + S'_i + (\mu_{\widetilde{q}_i} - m'_i)^2\right] \tag{19}$$

where $\sigma_{\widetilde{q}_i}^2$ is the variance of $\widetilde{q}(f_i)$. This function can be further simplified using the quantile based reformulation of $\mathrm{W}_2^2(\widetilde{q}_i, \mathcal{N}_i)$ (see details in Appendix F.4) which results in:

$$(19) = \mathrm{W}_2^2(\widetilde{q}_i, \mathcal{N}_i) + \|\boldsymbol{a}\|_2^2\sigma_{\widetilde{q}_i}^2 \underbrace{- 2^{\frac{3}{2}}\boldsymbol{a}^{\mathsf{T}}\boldsymbol{a}'c_{\widetilde{q}_i}S_i'^{\frac{1}{2}} + \|\boldsymbol{a}'\|_2^2 S'_i}_{(B)}. \tag{20}$$

Now, we are left with optimizing $m'_i$, $S'_i$ and $\boldsymbol{S}'_{\backslash ii}$. To optimize $\boldsymbol{S}'_{\backslash ii}$, which only exists in the above term $(B)$, we zero the derivative of $(B)$ w.r.t. $\boldsymbol{S}'_{\backslash ii}$ and this yields:

$$\boldsymbol{a}'^* = 2^{\frac{1}{2}}(S'_i)^{-\frac{1}{2}}c_{\widetilde{q}_i}\boldsymbol{a} \overset{\text{Eqn. (13)}}{\Longrightarrow} \boldsymbol{S}'^*_{\backslash ii} = (2S'_i)^{\frac{1}{2}}c_{\widetilde{q}_i}\boldsymbol{a}, \tag{21}$$

and the minimum value of Equation (20) is

$$\min_{\boldsymbol{S}'_{\backslash ii}}(20) = \mathrm{W}_2^2(\widetilde{q}_i, \mathcal{N}_i) + \|\boldsymbol{a}\|_2^2(\sigma_{\widetilde{q}_i}^2 - 2c_{\widetilde{q}_i}^2). \tag{22}$$

The results of optimizing $m'_i$ and $S'_i$ in the above equation have already been provided in Equation (5): $m'^*_i = \mu_{\widetilde{q}_i}$ and $S'^*_i = 2c_{\widetilde{q}_i}^2$. By plugging them into Equation (21) and Equation (18), we have

$\boldsymbol{a}'^* = \boldsymbol{a}$ and $\boldsymbol{b}'^* = \boldsymbol{b}$. Finally, using Equation (16), we obtain $q^{\backslash i}(\boldsymbol{f}_{\backslash i}|f_i) = \mathcal{N}(\boldsymbol{f}_{\backslash i}|\boldsymbol{a}f_i + \boldsymbol{b}, \boldsymbol{S}_{\backslash i|i}) = \mathcal{N}(\boldsymbol{f}_{\backslash i}|\boldsymbol{a}'f_i + \boldsymbol{b}', \boldsymbol{S}'_{\backslash i|i}) = \mathcal{N}(\boldsymbol{f}_{\backslash i}|f_i)$ , which concludes the proof. $\qquad\square$

## F.1  Details of Eqn. (6)

$$
\begin{aligned}
\mathrm{KL}(\widetilde{q}(\boldsymbol{f})\|\mathcal{N}(\boldsymbol{f})) &= \int \widetilde{q}(\boldsymbol{f}) \log \frac{\widetilde{q}(\boldsymbol{f}_{\backslash i}|f_i)\widetilde{q}(f_i)}{\mathcal{N}(\boldsymbol{f}_{\backslash i}|f_i)\mathcal{N}(f_i)} \, \mathrm{d}\boldsymbol{f} \\
&= \int \widetilde{q}(f_i) \log \frac{\widetilde{q}(f_i)}{\mathcal{N}(f_i)} \, \mathrm{d}f_i + \int \widetilde{q}(f_i) \int \widetilde{q}(\boldsymbol{f}_{\backslash i}|f_i) \log \frac{\widetilde{q}(\boldsymbol{f}_{\backslash i}|f_i)}{\mathcal{N}(\boldsymbol{f}_{\backslash i}|f_i)} \, \mathrm{d}\boldsymbol{f}_{\backslash i} \, \mathrm{d}f_i \\
&= \mathrm{KL}\big(\widetilde{q}(f_i)\|\mathcal{N}(f_i)\big) + \mathbb{E}_{\widetilde{q}(f_i)}\Big[\mathrm{KL}\big(\widetilde{q}(\boldsymbol{f}_{\backslash i}|f_i)\|\mathcal{N}(\boldsymbol{f}_{\backslash i}|f_i)\big)\Big]
\end{aligned}
$$

$$
\begin{aligned}
\widetilde{q}(\boldsymbol{f}_{\backslash i}|f_i) = \frac{\widetilde{q}(\boldsymbol{f})}{\widetilde{q}(f_i)} &\propto \frac{p(\boldsymbol{f})\cancel{p(y_i|f_i)}\prod_{j\neq i}t_j(\boldsymbol{f})}{q^{\backslash i}(f_i)\cancel{p(y_i|f_i)}} \\
&= q^{\backslash i}(\boldsymbol{f}_{\backslash i}|f_i).
\end{aligned}
\tag{23}
$$

## F.2  Details of Eqn. (10)

$$
\begin{aligned}
\mathrm{W}_2^2\left(\widetilde{q}(\boldsymbol{f}),\mathcal{N}(\boldsymbol{f})\right) &\equiv \inf_{\pi\in U(\widetilde{q},\mathcal{N})} \mathbb{E}_\pi\left(\|\boldsymbol{f}-\boldsymbol{f}'\|_2^2\right) \\
&= \inf_{\pi\in U(\widetilde{q},\mathcal{N})} \mathbb{E}_\pi\left(\|f_i-f_i'\|_2^2\right) + \mathbb{E}_\pi\left(\|\boldsymbol{f}_{\backslash i}-\boldsymbol{f}'_{\backslash i}\|_2^2\right) \\
&\overset{(a)}{=} \inf_{\pi\in U(\widetilde{q},\mathcal{N})} \mathbb{E}_{\pi_i}\left[\|f_i-f_i'\|_2^2 + \mathbb{E}_{\pi_{\backslash i|i}}\left(\|\boldsymbol{f}_{\backslash i}-\boldsymbol{f}'_{\backslash i}\|_2^2\right)\right] \\
&\overset{(b)}{=} \inf_{\pi_i} \mathbb{E}_{\pi_i}\left[\|f_i-f_i'\|_2^2 + \inf_{\pi_{\backslash i|i}} \mathbb{E}_{\pi_{\backslash i|i}}\left(\|\boldsymbol{f}_{\backslash i}-\boldsymbol{f}'_{\backslash i}\|_2^2\right)\right] \\
&= \inf_{\pi_i} \mathbb{E}_{\pi_i}\left[\|f_i-f_i'\|_2^2 + \mathrm{W}_2^2(\widetilde{q}_{\backslash i|i},\mathcal{N}_{\backslash i|i})\right] \\
&\overset{(c)}{=} \inf_{\pi_i} \mathbb{E}_{\pi_i}\left[\|f_i-f_i'\|_2^2 + \mathrm{W}_2^2(q^{\backslash i}_{\backslash i|i},\mathcal{N}_{\backslash i|i})\right],
\end{aligned}
$$

where the superscript prime indicates that the variable is from the Gaussian $\mathcal{N}$. In (a), $\pi_i = \pi(f_i, f_i')$ and $\pi_{\backslash i|i} = \pi(\boldsymbol{f}_{\backslash i}, \boldsymbol{f}'_{\backslash i}|f_i, f_i')$. In (b), the first and the second inf are over $U(\widetilde{q}_i, \mathcal{N}_i)$ and $U(\widetilde{q}_{\backslash i|i}, \mathcal{N}_{\backslash i|i})$ respectively. (c) is due to $\widetilde{q}(\boldsymbol{f}_{\backslash i}|f_i)$ being equal to $q^{\backslash i}(\boldsymbol{f}_{\backslash i}|f_i)$ (refer to Eqn. (23)).

## F.3  Details of Eqn. (19)

$$
\min_{\boldsymbol{m}'_{\backslash i}} \text{Eqn. (17)}
$$

$$
= \inf_{\pi_i} \mathbb{E}_{\pi_i}\left[\|f_i-f_i'\|_2^2 + \|\boldsymbol{a}(f_i-\mu_{\widetilde{q}_i}) - \boldsymbol{a}'(f_i'-m_i')\|_2^2\right]
$$

$$
= \inf_{\pi_i} \mathbb{E}_{\pi_i}\left[\|f_i-f_i'\|_2^2\right] + \|\boldsymbol{a}\|_2^2\sigma_{\widetilde{q}_i}^2 + \|\boldsymbol{a}'\|_2^2 S_i' - 2\boldsymbol{a}^\mathsf{T}\boldsymbol{a}'\mathbb{E}_{\pi_i}\left(f_i f_i' - \mu_{\widetilde{q}_i}m_i'\right)
$$

$$
= \inf_{\pi_i} \mathbb{E}_{\pi_i}\left[\|f_i-f_i'\|_2^2\right] + \|\boldsymbol{a}\|_2^2\sigma_{\widetilde{q}_i}^2 + \|\boldsymbol{a}'\|_2^2 S_i' + \boldsymbol{a}^\mathsf{T}\boldsymbol{a}'\mathbb{E}_{\pi_i}\left(\|f_i-f_i'\|_2^2 - f_i^2 - (f_i')^2 + 2\mu_{\widetilde{q}_i}m_i'\right)
$$

$$
= \inf_{\pi_i} \mathbb{E}_{\pi_i}\left[\|f_i-f_i'\|_2^2\right] + \|\boldsymbol{a}\|_2^2\sigma_{\widetilde{q}_i}^2 + \|\boldsymbol{a}'\|_2^2 S_i' + \boldsymbol{a}^\mathsf{T}\boldsymbol{a}'\mathbb{E}_{\pi_i}\Big(\|f_i-f_i'\|_2^2 - (f_i-\mu_{\widetilde{q}_i})^2 -
$$

$$
2 f_i\mu_{\widetilde{q}_i} + \mu_{\widetilde{q}_i}^2 - (f_i'-m_i')^2 - 2f_i'm_i' + (m_i')^2 + 2\mu_{\widetilde{q}_i}m_i'\Big)
$$

$$
= (1 + \boldsymbol{a}^\mathsf{T}\boldsymbol{a}')\mathrm{W}_2^2(\widetilde{q}_i,\mathcal{N}_i) + \|\boldsymbol{a}\|_2^2\sigma_{\widetilde{q}_i}^2 + \|\boldsymbol{a}'\|_2^2 S_i' - \boldsymbol{a}^\mathsf{T}\boldsymbol{a}'\left(\sigma_{\widetilde{q}_i}^2 + \mu_{\widetilde{q}_i}^2 + S_i' + (m_i')^2 - 2\mu_{\widetilde{q}_i}m_i'\right)
$$

$$
= (1 + \boldsymbol{a}^\mathsf{T}\boldsymbol{a}')\mathrm{W}_2^2(\widetilde{q}_i,\mathcal{N}_i) + \|\boldsymbol{a}\|_2^2\sigma_{\widetilde{q}_i}^2 + \|\boldsymbol{a}'\|_2^2 S_i' - \boldsymbol{a}^\mathsf{T}\boldsymbol{a}'\left[\sigma_{\widetilde{q}_i}^2 + S_i' + (\mu_{\widetilde{q}_i} - m_i')^2\right]
$$

### F.4    Details of Eqn. (19)

We first use Proposition 1 to reformulate the $L_2$ WD $W_2^2(\widetilde{q}_i, \mathcal{N}_i)$ as:

$$
\begin{aligned}
W_2^2(\widetilde{q}_i, \mathcal{N}_i) &= \int_0^1 \left( F_{\widetilde{q}_i}^{-1}(y) - m_i' - \sqrt{2S_i'}\mathrm{erf}^{-1}(2y-1) \right)^2 \mathrm{d}y, \\
&= \int_0^1 (F_{\widetilde{q}_i}^{-1}(y) - m_i')^2 + 2S_i'\mathrm{erf}^{-1}(2y-1)^2 - 2\sqrt{2S_i'}\mathrm{erf}^{-1}(2y-1)(F_{\widetilde{q}_i}^{-1}(y) - m_i') \, \mathrm{d}y, \\
\\
&= \int_0^1 (F_{\widetilde{q}_i}^{-1}(y) - \mu_{\widetilde{q}_i} + \mu_{\widetilde{q}_i} - m_i')^2 \, \mathrm{d}y + S_i' - 2\sqrt{2S_i'}c_{\widetilde{q}_i}, \\
&= \sigma_{\widetilde{q}_i}^2 + (\mu_{\widetilde{q}_i} - m_i')^2 + S_i' - 2c_{\widetilde{q}_i}\sqrt{2S_i'},
\end{aligned}
$$

where $F_{\widetilde{q}_i}^{-1}(y)$ is the quantile function of $\widetilde{q}(f_i)$ and $c_{\widetilde{q}_i} \equiv \int_0^1 F_{\widetilde{q}_i}^{-1}(y)\mathrm{erf}^{-1}(2y-1)\,\mathrm{d}y$. Next, we plug this reformulation into Eqn. (19):

$$
\text{Eqn. (19)} = W_2^2(\widetilde{q}_i, \mathcal{N}_i) + \boldsymbol{a}^\mathsf{T}\boldsymbol{a}'W_2^2(\widetilde{q}_i, \mathcal{N}_i) + \|\boldsymbol{a}\|_2^2\sigma_{\widetilde{q}_i}^2 + \|\boldsymbol{a}'\|_2^2 S_i' - \boldsymbol{a}^\mathsf{T}\boldsymbol{a}'\left[ \sigma_{\widetilde{q}_i}^2 + S_i' + (\mu_{\widetilde{q}_i} - m_i')^2 \right]
$$

$$
\begin{aligned}
&= W_2^2(\widetilde{q}_i, \mathcal{N}_i) + \boldsymbol{a}^\mathsf{T}\boldsymbol{a}'\left[ \cancel{\sigma_{\widetilde{q}_i}^2} + \cancel{(\mu_{\widetilde{q}_i} - m_i')^2} + \cancel{S_i'} - 2c_{\widetilde{q}_i}\sqrt{2S_i'} \right] + \|\boldsymbol{a}\|_2^2\sigma_{\widetilde{q}_i}^2 + \|\boldsymbol{a}'\|_2^2 S_i' \\
&\quad - \boldsymbol{a}^\mathsf{T}\boldsymbol{a}'\left[ \cancel{\sigma_{\widetilde{q}_i}^2 + S_i' + (\mu_{\widetilde{q}_i} - m_i')^2} \right] \\
&= W_2^2(\widetilde{q}_i, \mathcal{N}_i) - 2c_{\widetilde{q}_i}\sqrt{2S_i'}\boldsymbol{a}^\mathsf{T}\boldsymbol{a}' + \|\boldsymbol{a}\|_2^2\sigma_{\widetilde{q}_i}^2 + \|\boldsymbol{a}'\|_2^2 S_i'
\end{aligned}
$$

## G    More Details of EP

We use the expressions $\widetilde{q}(\boldsymbol{f}) = q^{\backslash i}(\boldsymbol{f})p(y_i|f_i)/Z_{\widetilde{q}}$ and $q^{\backslash i}(\boldsymbol{f}) = q(\boldsymbol{f})/(t_i(f_i)Z_{q^{\backslash i}})$, and the derivation of $\mathrm{KL}(\widetilde{q}(\boldsymbol{f})\|q(\boldsymbol{f})) = \mathrm{KL}(\widetilde{q}(f_i)\|q(f_i))$ is shown as below:

$$
\begin{aligned}
\mathrm{KL}(\widetilde{q}(\boldsymbol{f})\|q(\boldsymbol{f})) &= \int \widetilde{q}(\boldsymbol{f}) \log \frac{q^{\backslash i}(\boldsymbol{f})p(y_i|f_i)}{Z_{\widetilde{q}}q(\boldsymbol{f})} \, \mathrm{d}\boldsymbol{f} \\
&= \int \widetilde{q}(\boldsymbol{f}) \log \frac{\cancel{q(\boldsymbol{f})}p(y_i|f_i)}{Z_{q^{\backslash i}}Z_{\widetilde{q}}\cancel{q(\boldsymbol{f})}t_i(f_i)} \, \mathrm{d}\boldsymbol{f} \\
&= \int \widetilde{q}(f_i) \log \frac{p(y_i|f_i)}{Z_{q^{\backslash i}}Z_{\widetilde{q}}t_i(f_i)} \, \mathrm{d}f_i \\
&= \int \widetilde{q}(f_i) \log \frac{q^{\backslash i}(f_i)p(y_i|f_i)}{Z_{q^{\backslash i}}Z_{\widetilde{q}}q^{\backslash i}(f_i)t_i(f_i)} \, \mathrm{d}f_i \\
&= \int \widetilde{q}(f_i) \log \frac{\widetilde{q}(f_i)}{q(f_i)} \, \mathrm{d}f_i \\
&= \mathrm{KL}(\widetilde{q}(f_i)\|q(f_i))
\end{aligned}
$$

## H    Predictive Distributions of Poisson Regression

Given the approximate predictive distribution $f(\boldsymbol{x}_*) = \mathcal{N}(\mu_*, \sigma_*^2)$ and the relation $g(f) = f^2$, it is straightforward to derive the corresponding $g(\boldsymbol{x}_*) \sim \mathrm{Gamma}(k_*, c_*)^2$ where the shape $k_*$ and the scale $c_*$ are expressed as [56, 61]:

$$
k_* = \frac{(\mu_*^2 + \sigma_*^2)^2}{2\sigma_*^2(2\mu_*^2 + \sigma_*^2)}, \quad c_* = \frac{2\sigma_*^2(2\mu_*^2 + \sigma_*^2)}{\mu_*^2 + \sigma_*^2}.
$$

Furthermore, the predictive distribution of the count value $y \in \mathbb{N}$ can also be derived straightforwardly:

$$
\begin{aligned}
p(y) &= \int_0^\infty p(g_*)p(y|g_*) \, \mathrm{d}g_* \\
&= \int \mathrm{Gamma}(g_*|k_*, c_*)\mathrm{Poisson}(y|g_*) \, \mathrm{d}g_* \\
&= \frac{c_*^y(c_* + 1)^{-k_* - y}\Gamma(k_* + y)}{y!\Gamma(k_*)} = \mathrm{NB}(y|k_*, c_*/(1 + c_*)),
\end{aligned}
$$

where $g_* = g(\boldsymbol{x}_*)$ and NB denotes the negative binomial distribution. The mode is obtained as $\lfloor c_*(k_* - 1) \rfloor$ if $k_* > 1$ else 0.

## I  Proof of Corollary 2.2

Since the site approximations of both EP and QP are Gaussian, we may analyse the predictive variances using results from the regression with Gaussian likelihood function case, namely the well known Equation (3.61) in [47]:

$$
\sigma^2(f_*) = k(\boldsymbol{x}_*, \boldsymbol{x}_*) - \boldsymbol{k}_*^\mathsf{T}(K + \widetilde{\Sigma})^{-1}\boldsymbol{k}_*, \tag{24}
$$

where $f_* = f(\boldsymbol{x}_*)$ is the evaluation of the latent function at $\boldsymbol{x}_*$ and $\boldsymbol{k}_* = [k(\boldsymbol{x}_*, \boldsymbol{x}_1), \cdots, k(\boldsymbol{x}_*, \boldsymbol{x}_N)]^\mathsf{T}$ is the covariance vector between the test data $\boldsymbol{x}_*$ and the training data $\{\boldsymbol{x}_i\}_{i=1}^N$, $K$ is the prior covariance matrix and $\widetilde{\Sigma}$ is the diagonal matrix with elements of site variances $\widetilde{\sigma}_i^2$.

After updating the parameters of a site function $t_i(f_i)$, the term $(K + \widetilde{\Sigma})^{-1}$ is updated to $(K + \widetilde{\Sigma} + (\widetilde{\sigma}_{i,\text{new}}^2 - \widetilde{\sigma}_i^2)\boldsymbol{e}_i\boldsymbol{e}_i^\mathsf{T})^{-1}$ where $\widetilde{\sigma}_{i,\text{new}}$ is the site variance estimated by EP or QP and $\boldsymbol{e}_i$ is a unit vector in direction $i$. Using the Woodbury, Sherman & Morrison formula [47, A.9], we rewrite $(K + \widetilde{\Sigma} + (\widetilde{\sigma}_{i,\text{new}}^2 - \widetilde{\sigma}_i^2)\boldsymbol{e}_i\boldsymbol{e}_i^\mathsf{T})^{-1}$ as

$$
\begin{aligned}
&(K + \widetilde{\Sigma} + (\widetilde{\sigma}_{i,\text{new}}^2 - \widetilde{\sigma}_i^2)\boldsymbol{e}_i\boldsymbol{e}_i^\mathsf{T})^{-1} \\
&\equiv (A^{-1} + (\widetilde{\sigma}_{i,\text{new}}^2 - \widetilde{\sigma}_i^2)\boldsymbol{e}_i\boldsymbol{e}_i^\mathsf{T})^{-1} \\
&= A - A\boldsymbol{e}_i[(\widetilde{\sigma}_{i,\text{new}}^2 - \widetilde{\sigma}_i^2)^{-1} + \boldsymbol{e}_i^\mathsf{T}A\boldsymbol{e}_i]^{-1}\boldsymbol{e}_i^\mathsf{T}A \\
&\equiv A - \boldsymbol{s}_i[(\widetilde{\sigma}_{i,\text{new}}^2 - \widetilde{\sigma}_i^2)^{-1} + A_{ii}]^{-1}\boldsymbol{s}_i^\mathsf{T} \\
&= A - \frac{1}{(\widetilde{\sigma}_{i,\text{new}}^2 - \widetilde{\sigma}_i^2)^{-1} + A_{ii}}\boldsymbol{s}_i\boldsymbol{s}_i^\mathsf{T}
\end{aligned}
$$

where $A = (K + \widetilde{\Sigma})^{-1}$ and $\boldsymbol{s}_i$ is the $i$'th column of $A$. Putting the above expression into Equation (24), we have that the predictive variance is updated according to:

$$
\sigma_{\text{new}}^2(f_*) = k(\boldsymbol{x}_*, \boldsymbol{x}_*) - \boldsymbol{k}_*^\mathsf{T}A\boldsymbol{k}_* + \frac{1}{(\widetilde{\sigma}_{i,\text{new}}^2 - \widetilde{\sigma}_i^2)^{-1} + A_{ii}}\boldsymbol{k}_*^\mathsf{T}\boldsymbol{s}_i\boldsymbol{s}_i^\mathsf{T}\boldsymbol{k}_*.
$$

In EP and QP, the first two terms on the r.h.s. of the above equation are equivalent. As the site variance provided by QP is less or equal to that by EP, *i.e.*, $\widetilde{\sigma}_{i,\text{QP}}^2 \leq \widetilde{\sigma}_{i,\text{EP}}^2$, the third term on the r.h.s. for QP is less or equal to that for EP. Therefore, the predictive variance of QP is less or equal to that of EP: $\sigma_{\text{QP}}^2(f_*) \leq \sigma_{\text{EP}}^2(f_*)$.

## J  Lookup Tables

To speed up updating variances $\sigma_{\text{QP}}^2$ in QP, we pre-compute the integration in Equation (5) over a grid of cavity parameters $\mu$ and $\sigma$, and store the results into lookup tables. Consequently, each update step obtains $\sigma_{\text{QP}}^2$ simply based on the lookup tables. Concretely, for the GP binary classification, we compute Equation (5) with $\mu$, $\sigma$ and $y$ varying from -10 to 10, 0.1 to 10 and $\{-1, 1\}$ respectively. $\mu$ and $\sigma$ vary in a linear scale and a log10 scale respectively, and both have a step size of 0.001. The resulting lookup tables has a size of $20001 \times 2001$. In a similar way, we make the lookup table

**Algorithm 1** Expectation (Quantile) Propagation

---

**Input:** $p(\boldsymbol{f}), p(y_i|f_i), t_i(f_i), i = 1, \cdots, N, \boldsymbol{\theta}$
**Output:** $q(\boldsymbol{f})$          approximate posterior

 1: **repeat**
 2:      compute $q(\boldsymbol{f}) \propto p(\boldsymbol{f}) \prod_i t_i(f_i)$          by (1)
 3:      **repeat**
 4:         **for** $i = 1$ to $N$ **do**
 5:            compute $q^{\backslash i}(f_i) \propto q(f_i)/t_i(f_i)$          cavity
 6:            compute $\widetilde{q}(f_i) \propto q^{\backslash i}(f_i)p(y_i|f_i)$          tilted
 7:            **if** EP **then**
 8:               $t_i(f_i) \propto \text{proj}_{\text{KL}}[\widetilde{q}(f_i)]/q^{\backslash i}(f_i)$          by (3)(4)
 9:            **else if** QP **then**
10:               $t_i(f_i) \propto \text{proj}_{\text{W}}[\widetilde{q}(f_i)]/q^{\backslash i}(f_i)$          by (5)(4)
11:            **end if**
12:            update $q(\boldsymbol{f}) \propto p(\boldsymbol{f}) \prod_i t_i(f_i)$          by (1)
13:         **end for**
14:      **until** convergence
15:      $\boldsymbol{\theta} = \text{argmax}_{\boldsymbol{\theta}} \log q(\mathcal{D})$          by (2)
16: **until** convergence
17: **return** $q(\boldsymbol{f})$

---

for the Poisson regression. In the experiments, we exploit the linear interpolation to fit $\sigma^2_{\text{QP}}$ given $\mu \in [-10, 10]$ and $\sigma \in [0.1, 10]$, and if $\mu$ and $\sigma$ lie out of the lookup table, $\sigma^2_{\text{QP}}$ is approximately computed by the EP update formula, i.e., $\sigma^2_{\text{QP}} \approx \sigma^2_{\text{EP}}$. On Intel(R) Xeon(R) CPU E5-2680 v4 @ 2.40GHz, we observe the running time of EP and QP is almost the same.

## Footnotes

[2]$\mathrm{Gamma}(x|k, c) = \frac{1}{\Gamma(k)c^k}x^{k-1}e^{-x/c}$.