[Reviews · NeurIPS 2020]

Review 1

Summary and Contributions: This manuscript contributes an algorithm for approximate Gaussian process inference, for non-Gaussian likelihoods. It assumes the likelihood is a product of terms, each of which depends on the latent GP function at one point. The algorithm proposed here, Quantile Propagation (QP), is very similar to Expectation Propagation (EP). They both approximate the non-Gaussian likelihood terms with a Gaussian, and they both do so iteratively, by optimising each local likelihood according to some criterion. The difference is, EP minimises the forward KL(p||q) divergence for each local factor at each iteration, and QP minimises the L2-Wasserstein distance at each iteration. The authors prove that this can be done in an efficient way while being as correct as EP (the locality property), and provide a practical algorithm. The algorithm gives the approximate posterior the same mean as EP at each iteration, and a smaller variance. This is desirable, because EP usually overestimates the variance. Experiments on binary classification show that accuracy is almost exactly the same as EP, while predictive log-likelihood is better.

Strengths: The algorithm is conceptually a very small change from EP: merely change the metric to be optimised for every local factor from forward KL-divergence, to L2-Wasserstein. This makes it easy to understand for people who already work in this field. The algorithm should be straightforward to extend with some of the EP extensions like Stochastic EP, and Sparse EP [15, BYT-2017], and probably even multi-class classification in which each training example has one site function per class [VCHL-2017]. The forward-KL from EP is known to overestimate uncertainty. I think replacing it by the L2-Wasserstein should strike the majority of researchers as an obviously desirable improvement. The Wasserstein distance is very hard to calculate. It is even harder to do approximate inference with it. A general procedure for approximate Bayesian inference by minimising some sort of Wasserstein distance to the posterior would be a large boon for the field. The steps taken by this paper are encouraging. The paper proves that the approximations the algorithm makes are, in a sense, "not worse" than EP. That is: the authors don't prove QP converges, but EP isn't proven to converge either. The authors do prove that minimising L2-W distance to the tilted distribution *can* be done by 1-d updates, which is the crucial reason this algorithm is roughly as fast as EP. [VCHL-2017]: https://arxiv.org/abs/1706.07258

Weaknesses: After reading the rebuttals and reviewer discussion, I realise that I was wrong about EP overestimating the variance and the strength of the paper's empirical results, so I have decided to downgrade my score. I still believe this paper should be accepted, but I'm less confident of the matter. Here are the things I changed my mind about, to more critical: - Does EP really overestimate the posterior variance? EP should overestimate the *support* of distributions, because the forward-KL covers all modes with a (unimodal) Gaussian. But this does not necessarily imply that the variance is overestimated, and locally the variance is matched exactly. The Power EP results by Bui, Yan and Turner (2016, https://arxiv.org/abs/1605.07066), suggest that interpolating between EP and VI (setting alpha=0.5) gives better performance, but the variance is not discussed much. I think the existing evidence that I know of is insufficient to support this claim, and the authors should support it. Another reviewer pointed out during the discussion that "MCMC experiments have shown that EP with KL is surprisingly accurate. The paper lacks a comparison in this respect". The references they gave are "[2, Figures 4+5] and the follow up work [3, Figure 9]". - Empirical results: I explicitly calculated the p-values for the 2-sample t-tests between EP and QP, in the test log-likelihood results. With sample size = 100, and the code listing below [1], the p-values are all very high: the lowest, for Wine2, is 0.24. The experiments need to be run more times to distinguish whether QP really estimates the test log-likelihood better. I'm not sure how the asterisk * "more than 90% of QP are better than EP" was calculated, it doesn't fit; unless the experiments were sorted by performance before calculating it. I'm not picking on the test error because it's not very surprising that it is pretty much the same as EP, because the mean updates are the same (even if the variance updates are a bit different). - No marginal likelihood experiments: this is the crucial quantity used to compare Bayesian models of any kind. All other GP approximation schemes that are used can estimate it, worse or better. EP tends to overestimate the likelihood (see Bui, Yan and Turner again), so if QP overestimates it less it is a clear win. [1] Code: import math, scipy.stats def t(m1, s1, m2, s2, n=100.): sp_n = math.sqrt((s1**2 + s2**2) / n) t_stat = (m1 - m2) / sp_n df = 2*n - 2 print(df, t_stat) cdf = scipy.stats.t.cdf(x=t_stat, df=df, loc=0., scale=1.) return 2 * min(cdf, 1-cdf) [2] https://www.jmlr.org/papers/volume6/kuss05a/kuss05a.pdf [3] https://www.jmlr.org/papers/volume9/nickisch08a/nickisch08a.pdf ==== The paper is technically very strong and very well-written. It has very few weaknesses. Maybe it needs a bit more signposting for the reader, which I'll write later on. For the future, there are some reasons to be pessimistic about this line of inquiry: EP requires an analytic expression for the 1st and 2nd moments of the local factors. QP needs to minimise the L2-Wasserstein distance, which involves an integral and two minimisation problems. It is plausible that these are found analytically for fewer distributions than EP, particularly for likelihoods that need to account for more than one latent function (like the Softmax). Though, in fairness, EP also cannot handle the Softmax likelihood. The "Broader Impact" statement is weak. Indeed, you don't expect this algorithm to have any immediate effects in society. But won't it impact the broader field of GPs? of approximate Bayesian inference? These things should also go in the section (eg. see unofficial guide, https://medium.com/@GovAI/a-guide-to-writing-the-neurips-impact-statement-4293b723f832 ). I practically wrote the impact statement for this paper in the "Strengths of the paper" box.

Correctness: I haven't checked all details of the calculations in the proofs, but they seem roughly correct. The empirical comparison is fair and sound, but: what do the bolded entries mean in Table 1? Which algorithm comes on top in statistically significant comparisons? (from what I can tell, dividing the standard deviations after the \pm sign by 100, says that the answer is "yes").

Clarity: Very much so. Some small issues: - Corollary 2.2 should read "The predictive variances [...] satisfy: [...] *after one update of QP or EP*", or similar. As the authors state, they haven't proven that the variance of QP is smaller at convergence (though it likely can be proven). - line 212: theorem 2, not theorem E. In my view, though I may be wrong, the importance of the theorems in section 5 is that they show that the 1-dimensional procedure in section 4 is correct, in that it minimises the L2-W to the tilted distributions. Thus, I suggest that the authors could state so around line ~230.

Relation to Prior Work: Yes, in section 2.

Reproducibility: Yes

Additional Feedback:


Review 2

Summary and Contributions: This paper describes a variant of expectation propagation based on minimizing the Wasserstein distance between probability distributions, instead of the typical KL divergence. The method is validated in the context of Gaussian process binary classification and Gaussian processes with Poisson likelihood factors. The results show small improvements in the test log-likelihood.

Strengths: - Well written paper. - Simple approach. - Improvements over expectation propagation.

Weaknesses: - The improvements are fairly small. - The method is probably more expensive than expectation propagation. - It is not clear how hyper-parameter optimization is done. See below.

Correctness: As far as I have checked the paper is correct.

Clarity: The paper is clearly written.

Relation to Prior Work: Related work is correctly cited and described. However, I have missed this reference: Hernández-Lobato, D., & Hernández-Lobato, J. M. (2016, May). Scalable gaussian process classification via expectation propagation. In Artificial Intelligence and Statistics (pp. 168-176).

Reproducibility: No

Additional Feedback: No code is given to reproduce the experiments found in the paper. Overall I believe that this is an interesting paper for the community working on Gaussian processes and approximate inference. However, I have some concerns with it. The experimental results show only small improvements. Therefore it is not clear if the proposed method is worth it. In particular, it is very likely that it is more expensive than EP. Therefore, I have missed extra experiments comparing both methods in terms of performance vs computational time. It is not clear how the estimate of the marginal likelihood is optimized. In particular, in EP one can show that at convergence, the parameters of the approximate factors can be considered fixed. That enables easy optimization and gradient computation of the estimate of the marginal likelihood. When the Wasserstein distance is used, it is not clear if these conditions are also met. Therefore, the optimization of the estimate of the marginal likelihood is questionable. I have also missed experimental results regarding the accuracy of the estimate of the marginal likelihood. The name quantile propagation is also questionable since the quantiles are not matched exactly, but approximately. I have read the authors' response. I have chosen to keep my score as it is.


Review 3

Summary and Contributions: This paper considers the problem of approximating a Bayesian model. In particular, the set-up includes a Gaussian process model, assumes a factorized form for the likelihood, and approximates the individual likelihoods with Gaussians, called site functions. This has been considered before, and the parameters of these individual site functions are optimized using Expectation Propagation, which projects the tilted distribution on the space of Gaussians. This projection step takes place in KL and results in convenient moment matching. The novel work of this paper considers replacing the KL projection with a Wasserstein projection since the Wasserstein distance defines a metric on the space of probability measures that have nice properties. In a similar way to the KL projection, this new Wasserstein projection results in 1D updates to the parameters of the site functions, and these parameters can be found by matching quantiles instead of moments, and the method is referred to as Quantile Propagation. Some theory is given showing that the variance of QP is upper bounded by that of EP, although no analysis of the fixed points of the methods is given. Experiments on a few benchmark datasets show a slight advantage for using QP over EP.

Strengths: - Use of the Wasserstein distance over divergences is preferred to deal with differing supports. - There is an advantage for using QP over EP in terms of smaller variance for the same cavity distribution. - The use of QP maintains the locality property, which means that the updates can be efficiently done in 1D. The numerical integration needed can be done efficiently in advance by making lookup tables offline.

Weaknesses: - Perhaps most importantly, empirical advantages of the method are not demonstrated. Together with the lack of theoretical motivation, this is the largest shortcoming of the work. I am left still not knowing what the primary motivation and reason for pursuing QP over EP is. - Lack of theoretical motivation. The main theory of the work relates to some variance bounds for QP versus EP, although these are just bounds. No gap between these methods is given. - The bounds only apply to when the same cavity distribution is used. No analysis of fixed points is given. - The algorithm is not so clearly explained in the main text, and is given in the appendix. - No convergence properties of the algorithm are discussed. - In cases where one does not have access to the CDF of the cavity distribution, the authors do not discuss performance. - The look-up tables depend on the tilted distribution and so change from problem to problem. - Showing convergence is still a major issue for the algorithm. Are there any modifications for future work for which you could prove convergence and analyze the fixed points? - Is there any guidance for constructing the look-up tables?

Correctness: The method seems to be correct and the empirical methodology is standard in the Bayesian setting.

Clarity: The paper is clear and well written in general.

Relation to Prior Work: The authors clearly tie the method in with the existing work on EP, and discuss recent extensions of EP.

Reproducibility: Yes

Additional Feedback: Overall, this work seems like a natural and nice extension of EP using a different metric. It is good to see that the locality property is maintained, and through look-up tables one could hope to have efficient algorithms. However, I am left wanting more - the primary issue being that it is not really demonstrated what the advantages of the proposed method are. There is some theory hinting at lower variance (although this is not quantitatively stated), but this is not demonstrated in the experiments - the error bars seem to be overlapping and the results do not seem to be significantly different. It is also not demonstrated how the method converges when compared with EP. #### POST REBUTTAL #### After reading the authors rebuttal, I still think the paper is marginally above the acceptance threshold. While the authors do show that the test log-likelihoods are consistently lower (in mean), they are not significantly so, and in fact the error bars given overlap in all cases. Therefore, the numerical experiments are not sufficiently convincing as a replacement for EP. The theory doesn't help either, since it gives the variance as a lower bound but offers no quantitative difference. On the other hand, I do agree that using the Wasserstein distance is desirable, and the work is interesting and novel.


Review 4

Summary and Contributions: The paper proposes an iterative message passing approach for approximate inference in GP models very similar to EP. The main difference to EP is that instead of th KL divergence, the Wasserstein distance is minimized in each site update step which results in a different update for the variance parameter as compare to EP while the mean update is exactly the same. Experiments on classification and Poisson regression illustrate aspects of the algorithm in practice.

Strengths: - The manuscript proposes to use the Wasserstein distance in EP, a divergence measure that gained attention recently but seemingly has not been used for GP inference before. - The authors provide some theoretical analysis allowing to gain some intuition on the EP procedure in particular the analysis of the variances is very intersting.

Weaknesses: - Note sure whether the authors intend to release code also upon acceptance but the statement in line 270 is a little unclear. If code is only available during the review phase, this is a clear minus. - The degree of novelty is pretty small as the framework is well known and only a tiny aspect is changed. - The paper contains a lot of known material on the one hand but has a lot of references to the Appendix which makes the paper a little hard to digest. I would suggest to remove textbook material on EP in favor of including some more material on the Wasserstein distance. - That said, I'm not sure whether the page on the locality property is enlightning and really surprising. This could in principle be part of the Appendix and leave more space for an algorithmic discussion of the required computations for the variance update. - EP suffers from stability problems when the moment updates are not numerically accurate e.g. as a result of quadrature approximations. I'm missing a discussion on the numerical aspects of the L2 Wasserstein distance computations. - I'm missing a discussion on the marginal likelihood and its accuracy. - I'm missing a discussion of whether and how further derivatives of the site update can be computed in order to perform marginal likelihood hyperparameter optimization. - I'm missing a discussion why values for p different from 2 are not interesting to consider. - The manuscript does not provide evidence whether the proposed divergence measure is better suited in cases where EP has "deficiencies" according to the authors. MCMC experiments have shown that EP with KL is surprisingly accurate. The paper lacks a comparison in this respect. The missing convergence proof for EP is clearly an issue but the 2nd and 3rd paragraph seem as if EP is a buggy approach per se. Please provide concise and concrete examples where EP with KL is problematic and demonstrate that EP with WD is any better.

Correctness: The theoretical results are most likely correct.

Clarity: The paper is mostly well written but a lot of the standard EP formulas can be ommited as they are text book material. Also a formulation in natural parameters would yield less cluttered expressions. Some intuition on the divergence measure along the lines of Figures 1+2 from [34] would help clarify the intuition.

Relation to Prior Work: Prior work is properly mentioned also the "divergence measure and message passing" framework of [34]. However, the paper could more clearly state that all they change compared to [34] is the local divergence measure. In particular, the Abstract and also the Introduction suggest that QP is a method different from EP and that EP and QP need to be contrasted. In fact QP is just a variant of EP with WD as divergence measure.

Reproducibility: Yes

Additional Feedback: - The second sentence of the Abstract is a pretty bold statement given that the authors choose p=2 for "computational efficiency" (line 185). Please modify. - Theorem 1 leaves somewhat open whether the distance is jointly convex in the vector (mu,sigma) or in some other parametrisation. - References, capitalization: Gaussian, Bayesian, Poisson

[Author Response · NeurIPS 2020]

Thanks to all of the reviewers for their time and effort, and both constructive and critical comments.

▷ **All Reviewers.** General Comment 1 - code and reproducibility Due to file size limits we included our code as an anonymous downloadable link in Appendix J, but we need to make this more obvious, thanks. We gladly commit to making the code publicly available on paper acceptance. General Comment 2.1 - Marginal Likelihood (ML) approximation Our ML estimate follows the standard approach in EP, which is to assume that the site approximations are fixed. Proving that "at convergence, the parameters of the approximate factors can be considered fixed" may require analyzing the energy function of our model and a good idea for future work, thanks. General Comment 2.2 - estimate of ML approximation We use the ML of the approximate GP in our work and also in EP. A previous study on GP classification [Assessing Approximate Inference for Binary Gaussian Process Classification, Kuss & Rasmussen] thoroughly compared various ML approximations, and found that the ML approximation we use matches the predictive accuracy very well. General Comment 3 - novelty We thank Reviewer 1 and Reviewer 3 for acknowledging the non-triviality of the locality property and practical efficiency of our work. Locality is our central result, and guarantees that the complex nested optimization of the Wasserstein distance reduces to a relatively simple and efficient 1-d update. Reviewer 1 puts it well, that "replacing it [KL] by the L2 Wasserstein should strike the majority of researchers as an obvious desirable improvement", and moreover that "Wasserstein distance is very hard to calculate. It is even harder to do approximate inference with it. A general procedure for approximate Bayesian inference by minimising some sort of Wasserstein distance to the posterior would be a large boon for the field." General Comment 4 - numerical stability If the CDF is accessible, Equation (5), which forms the basis of our lookup tables, is stable and it avoids divergence. Intuitively, this is because the Gaussian function in the integration is bounded by 1 and when the integration variable is very large or small, the Gaussian function approaches 0 rapidly. If the CDF isn't accessible, there are double integrations which can be computed in one pass. In such cases, the numerical stability may be problematic; investigating this would fairly deserve another paper.

▷ **Reviewer 1.** We thank reviewer 1 for their supportive comments and helpful suggestions on *e.g.* the broader impact, which we will incorporate in the final version. "Found analytically for fewer distributions than EP" While we agree that the number of analytically tractable cases for QP is probably less than that of EP, various numerical schemes may be employed and in many cases made efficient using lookup tables; see also General Comment 4.

▷ **Reviewer 2.** "No code is given" Please see General Comment 1. "not clear if the proposed method is worth it" Please see General Comment 3. "not clear how the estimate of the marginal likelihood is optimized"; "in EP one can show that at convergence, the parameters of the approximate factors can be considered fixed"; "the accuracy of the estimate of the marginal likelihood". Please see General Comments 2.1 and 2.2.

▷ **Reviewer 3.** "empirical advantages of the method are not demonstrated" We respectfully remind that in 8 out of 10 tasks, our method achieves consistently better test log-likelihoods on repeated experiments, than the strong baseline that is EP. Besides, Figure 1.a and 1.b illustrate the effectiveness of our method in alleviating the over-estimation of variances of EP. "primary motivation and reason for pursuing QP over EP" Please see General Comment 3. "No analysis of fixed points is given" Theoretical analysis of fixed points is an interesting area for future work, thanks. At present we offer an empirical analysis of fixed points as given in our experiment section. "any modifications for future work for which you could prove convergence and analyze the fixed points?" Changes such as the double loop EP [Expectation Consistent Approximate Inference, Opper & Winther] and proving the property pointed out by Reviewer 2 that "at convergence, the parameters of the approximate factors can be considered fixed" are an interesting direction for future work, thanks. However, we cannot yet rule out that our method is *already* provably convergent under appropriate assumptions. "Is there any guidance for constructing the look-up tables" Please see General Comment 1.

▷ **Reviewer 4.** "a lot of references to the Appendix ... the paper a little hard to digest" We will take the suggestion about bringing part of the supplement to the main paper, *e.g.* with regards to more explanation and discussion on WD. We will also use the extra page in the final version for this. "I'm not sure whether the page on the locality property is enlightening and really surprising" We respectfully disagree and are deeply disappointed that the reviewer places this comment in the list of 'weaknesses'. Locality is central to our contribution, and much harder to show here than for EP. We ask the reviewer to kindly consider the broader relevance outlined in General Comment 3. "Note sure whether the authors intend to release code" Please see General Comment 1. "The degree of novelty is pretty small" Please see General Comment 3. "marginal likelihood and its accuracy" Please see General Comments 2.1 and 2.2. "numerical aspects of the L2 Wasserstein distance computations" Please see General Comment 4. "A discussion why values for $p$ different from 2 are not interesting to consider." We briefly mention $p$ different from 2 in *e.g.* line 72–73, 188–189 and Appendix B. These cases are interesting but also *even more* challenging to handle. "does not provide evidence ... better suited in cases where EP has 'deficiencies'" The specific shortcoming of EP is over-estimation of variances as pointed out on *e.g.* lines 13, 29-31; we will clarify this even further in the final version, thanks. We give both theoretical analysis of local updates mitigating over-estimation of variances (sec. 4.3) and empirical evidence that at convergence our method predicts better and with lower predictive variances.

[Meta-Review · NeurIPS 2020]

As all reviewers agreed this is a clear novel contribution that proposes the Quantile Propagation (QP) algorithm, which operates similarly to Expectation Propagation (EP). While EP minimises the forward KL(p||q) divergence for each local factor, QP minimises the L2-Wasserstein distance. The authors have shown some theoretical results that QP can provide smaller variances than EP, which could be often beneficial in the reported experiments in Gaussian process classification. The use the Wasserstein distance for inference in GPs is interesting and thus the paper can be accepted on the basis of novelty. However, the claims of improved performance compared to EP are not really well supported experimentally. In fact, as the reviewers pointed out the new algorithm does not provided significantly better results than EP. Thus, the authors should down-weight their claims about "replacing EP". It would be useful also to add in the paper a clear and illustrative example pointing out the different behaviours of QP and EP.